

# Identifying sediment transport mechanisms from grain size-shape distributions

Johannes Albert van Hateren[1], Unze van Buuren[1], Sebastiaan Martinus Arens[2], Ronald Theodorus van Balen[1,3], Maarten Arnoud Prins[1]

[1]Faculty of Science, Department of Earth Sciences, Vrije Universiteit, Amsterdam, 1081 HV, The Netherlands

[2]Bureau for Beach and Dune Research, Soest, The Netherlands

[3]TNO-Geological Survey of the Netherlands, Utrecht, 3584 CB, The Netherlands

*Correspondence to:* Hans van Hateren (j.a.van.hateren@vu.nl)

**Abstract.** The way in which sediment is transported (creep, saltation, suspension), is traditionally interpreted from grain size distribution characteristics. However, the grain size range associated with transitions from one transport mode to the other is highly variable because it depends on the amount of transport energy available. In this study we present a novel methodology for determination of the sediment transport mode based on grain size and shape data from dynamic image analysis. The data are integrated into grain size-shape distributions and primary components are determined using end-member modelling. In

real-world datasets, primary components can be interpreted in terms of different transport mechanisms and/or sediment sources. Accuracy of the method is assessed using artificial datasets with known primary components that are mixed in known proportions. The results show that the proposed technique accurately identifies primary components with the exception of those primary components that only form minor contributions to the samples (highly mixed components).

The new method is also tested on sediment samples from an active aeolian system in the Dutch coastal dunes. Aeolian transport processes and geomorphology of these type of systems are well known and can therefore be linked to the spatial distribution of end members to assess the physical significance of the method's output. The grain size-shape distributions of the dune dataset are unmixed into three primary components. The spatial distribution of these components is constrained by geomorphology and reflects the three dominant aeolian transport processes known to occur along a beach-dune transect:

bedload on the beach and in notches that were dug by man through the shore-parallel foredune ridge, modified saltation on the windward and leeward slope of the intact foredune, and suspension in the vegetated hinterland. The three transport modes are characterised by distinctly different trends in grain shape with grain size: with increasing size, bedload shows a constant grain regularity, modified saltation a minor decrease in grain regularity and suspension a strong decrease in grain regularity. These trends, or in other words, the shape of the grain size-shape distributions, can be used to determine the transport mode

responsible for a sediment deposit. Results of the method are therefore less ambiguous than those of traditional grain-size distribution end-member modelling, especially if multiple transport modes occur or if primary components overlap in terms of grain size but differ in grain shape.





## 1 Introduction

Clastic sediment records are generally complex mixtures of grains due to variability in provenance, conditions in the source
and sink areas (climate, tectonics) and sorting during entrainment, transport and deposition. One of the greatest challenges in
sedimentology is to reconstruct signals of climate, tectonics and provenance from the sedimentary record (e.g. Garzanti et al.,
2007; Métivier et al., 1998; Prins and Weltje, 1999a; Zhang et al., 2016). These reconstructions are improved when the mixed
sedimentary record is unmixed into its primary constituent components (Weltje and Prins, 2007), a procedure which is also
termed end-member modelling. Various end-member modelling algorithms are used in sedimentology (e.g. Dietze et al., 2012;
Heslop et al., 2007; Paterson and Heslop, 2015; Weltje, 1997; Yu et al., 2016; Zhang et al., 2018). Although the algorithms
are capable of unmixing different types of data, they are commonly used on grain size distribution data (e.g. Dietze et al., 2014;
Liu et al., 2016; Stuut et al., 2002) and mineralogical data (Itambi et al., 2009; Weltje, 1995).

There are however at least two issues that complicate inferences based on single-property (size or mineralogy) end-member
modelling. First, sediment behaviour during uptake, transport and deposition is dictated by three grain properties: size, shape
and density (mineralogy) (Winkelmolen, 1971). Therefore, single-property end-member modelling results are prone to noise
from variability in the other two grain properties. The second issue is that the characterisation of sediment transport modes by
their grain size-distribution alone produces ambiguous results: the grain size range associated with the transitions between
transport modes (surface creep, saltation and suspension) depends on the amount of transport energy available and is therefore
highly variable (Visher, 1969). However, accurate identification of the transport mode is essential to a valid interpretation of
sedimentary records since the transport modes sort sediment grains differently during transport and are associated with
different transport velocities and distances.

In addition to sorting on grain size, sediment transport modes also sort shape in different ways. Studies on the influence of
particle shape on surface creep are sparse. Eisma (1965) inferred that it is likely that surface creep favours spherical grains
because these roll more easily. There are contradicting views regarding shape sorting during saltation: spherical grains bounce
higher (Eisma, 1965) and further (McCarthy and Huddle, 1938) and thus travel faster than non-spherical grains. However, they
are also more difficult to entrain (Winkelmolen, 1971). Likewise, studies on shape sorting in saltating transport under natural
conditions obtained contradictive results: some publications observed an increase in sphericity with transport distance
(MacCarthy and Huddle, 1938; Mazzullo et al., 1986), others a decrease (Eisma, 1965; Winkelmolen, 1971). This is further
complicated by the fact that inter-grain collision during (aeolian) saltation effectively rounds grains over longer distances
(Kuenen, 1960). During transport in suspension, settling velocity is the dominant sorting parameter (McCave, 2008; Pye,

1994). Settling velocity is higher for more spherical and regularly-shaped grains (e.g. Dietrich, 1982; Komar and Reimers, 1978; Wadell, 1934). Hence, in a suspended population of grains, larger grains are expected to be more irregularly shaped than smaller ones to remain below the fall velocity threshold for suspended transport. For example, Shang et al. (2018) observed that elongation increases with increasing size in Chinese loess. This decrease in grain regularity with increasing size should

lead to a characteristic size-shape trend of suspended sediment that is different from that of sediment transported as bedload; using grain shape in addition to grain size is therefore a promising approach to determine transport modes with less ambiguity.

In this study we outline a new method for determination of sediment transport processes involving 1) the integration of grain size and shape data into size-shape distributions and 2) end-member modelling on these distributions. To determine the

accuracy of the method, it is first tested on artificial grain size-shape datasets with known end members and known end-member mixing proportions. Subsequently, the method is applied to an active aeolian system in the Dutch coastal dunes (Ruessink et al., 2018). Aeolian transport processes and geomorphology of these type of systems are relatively well constrained (Arens et al., 2002) and can therefore be linked to the spatial distribution of end members to assess the physical significance of the method's output. The real-world dataset is also used to compare results of unmixing of size-shape distributions to results

of traditional unmixing based on grain-size distributions.

## 2 Material and methods

### 2.1 Dune dataset

The fieldwork area for our dataset is situated south of the town IJmuiden in a coastal dune region named National Park Zuid-Kennemerland (Appendix A1 and A2). In 2013, five notches were dug through the shore-parallel foredune ridge to promote

aeolian activity and dune migration (Appendix A2). The notches are roughly orientated along the dominant wind direction: west-southwest to east-northeast. Parabolic dunes have developed at the downwind end of the notches and large volumes of sand have been blown land-inward. From 2013 to 2016, approximately $87*10^3$ m$^3$ of sand was transported land-inward, 55% of which derived from the beach and 45% from erosion of the notches (Ruessink et al., 2018). Further land inward, vegetation has been removed from fossil parabolic dunes to stimulate reactivation of dunes (Appendix A2).

In order to assess the physical meaning of results from the new method, we divided the study area into its five main geomorphic features (Appendix A3): 1) The beach, which acts as a sediment source for aeolian transport when dry. 2) The foredune, on which marram grass (partly) impedes bedload transport. Near the crest, aeolian suspension and modified saltation are stimulated through increased wind velocities and high turbulence (Arens et al., 2002). 3) The notches, which enable bedload

transport towards the sand lobes that prograde into the vegetated hinterland (Ruessink et al., 2018). 4) The vegetated hinterland, where lower wind velocities and vegetation prevent bedload transport (Arens et al., 2002). And 5), the parabolic dunes that





were reactivated by removal of the vegetation cover. These dunes may form an additional source for the sediment flux in the hinterland (Arens et al., 2013).

In April 2017, shallow surface samples were obtained from one of the bare notches (n = 12) and from an undisturbed part of the foredune ridge (n= 18) (Appendix A2). Based on available flux data from sediment traps (not shown here), deposition rates land inward from sediment-trap row A (or perhaps B) are insufficient to sample recently transported material from the surface (Appendix A2). Samples from sediment traps (n = 23) are therefore used to study the land inward area. The traps are based on a design by Leatherman (1978) and consist of an 80 cm pvc pipe with a mid-height of approximately 1.5 m above ground level (Appendix B). Their opening is oriented into the dominant southwestern wind direction. At the back of the pipe a mesh with openings of 106 μm lets air and smaller particles through while trapping particles larger than 106 μm. Three time intervals characterised by high flux rates ('storm events') were sampled from the sediment traps (Table 1). Together, the sediment trap samples and surface samples form the dune dataset (Van Hateren et al., 2019).

Table 1. Wind conditions and sampling periods of the sediment trap samples. Sediment trap names are in reference to Appendix A2. Meteorological data were obtained from weather station IJmuiden, 3.5 km north of the fieldwork area.

| Sampling period | Number of samples | Sediment traps that were sampled | Mean daily wind speed (m/s) | Maximum daily wind speed (ms$^{-1}$) | Vector averaged wind direction (degrees) |
|---|---|---|---|---|---|
| 27/10/2015 - 17/11/2015 | 4 | A1, B1, C3 D3 | 8.5 | 15.5 | 214 |
| 17-11-2015 - 01-12-2015 | 15 | All | 11.8 | 17.1 | 255 |
| 01-12-2015 - 15-12-2015 | 4 | A1, B1, C3, D2 | 10.2 | 17.5 | 215 |

## 2.2 Dynamic image analysis

Sediment samples of approximately 2 grams are pre-treated with 5 ml $H_2O_2$ to remove organics, 5 ml HCl (10 ml if shell fragments are abundant) to remove carbonates and 300 mg $Na_4P_2O_7$ ·$10H_2O$ to disperse charged particles (Konert and Vandenberghe, 1997). Size and shape data are based on images of the grains obtained using a Sympatec Qicpic dynamic image analyser (Fig. 1A). The image analyser is set-up using a cuvette with 2 mm aperture. Pre-treated samples are sieved through a 1.6 mm mesh to protect the glass walls of this cuvette, thus limiting the maximum measurable grain size to 1600 μm. This is not of concern for the dune sands studied here, which show a maximum grain size of approximately 700 μm. The sediment samples are subsequently suspended in degassed water using a stirrer and pumped repeatedly through the cuvette for 10 minutes while being filmed at 25 frames per second, resulting in 15 thousand frames per sample. The frames measure 1024 by 1024 pixels with a pixel size of approximately 5 μm.

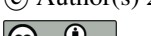



Image processing is carried out using a Matlab script written by the first author, for which Appendix C shows a workflow diagram. The particle size and shape characteristics that form the output of this script are described in Table 2 and an example is given in Fig. 1B. It is important to note that the major and minor grain diameters are based on the diameters of a fitted ellipse. These diameters are less sensitive to small scale particle roughness than the traditional Feret diameters (Feret, 1930). For the same reason, "the" diameter of the particle is given in the robust form of an area equivalent diameter (Table 2). We use ranges of interest in the graphs of size and shape distributions to focus on those size and shape classes that contain significant amounts of volume for the given dataset (Table 2).

Table 2. Summary of particle characteristics and derived size and shape variables. The table shows lower and upper limits for the variables as well as the size of the respective size or shape classes. The range of interest designates the range over which the sediments studied here contain significant volume for a given variable. The φ unit refers to Krumbein's log base 2 grain-size scale (Krumbein, 1938).





| Variable | Name | Description | Equation | Lower limit | Upper limit | Size | Number of classes | Range interval |
|---|---|---|---|---|---|---|---|---|
| $P_p$ | Perimeter | Length along particle boundary | - | - | - | - | - | - |
| $P_{ch}$ | Convex hull | Length along convex points on boundary | | | | | | |
| A | Area | Surface area of the particle | - | - | - | - | - | - |
| $D_A$ | Major diameter | Major diameter of ellipse fitted to particle | - | - | - | - | - | - |
| $D_B$ | Minor diameter | Minor diameter of fitted ellipse | - | - | - | - | - | - |
| D2d | Area equivalent diameter | Diameter of circle with area equal to A | $2\sqrt{\dfrac{A}{\pi}}$ | 13 µm | 2828 µm | $\frac{1}{8}\varphi$ | 62 | 105-70 |
| Con | Convexity | Ratio between convex hull length and perimeter length | $\dfrac{P_{ch}}{P_p}$ | 0 | 1 | 0.01 | 100 | 0.8 |
| Cc | Cox circularity (Cox, 1927) | Ratio that describes extent to which the area of a particle approximates that of a circle with the same perimeter | $4\pi\dfrac{A}{P_p^2}$ | 0 | 1 | 0.01 | 100 | 0.4 |
| Ar | Aspect ratio | Ratio of the major and minor diameter | $\dfrac{D_B}{D_A}$ | 0 | 1 | 0.01 | 100 | 0.3 |
| $V_A$ | - | Volume approximated from A, assuming a spherical particle shape | $\dfrac{4}{3}\pi^{-0.5}A^{1.5}$ | - | - | - | - | - |

## 2.3 Construction and unmixing of size-shape distributions

We explore the applicability of three shape parameters that are known to affect particle transport behaviour: convexity, Cox circularity (Cox, 1927) and aspect ratio (Table 2; Beal and Shepard, 1956; Dietrich, 1982; MacCarthy and Huddle, 1938;

5 Winkelmolen, 1971; Shang et al., 2018). These parameters relate to different aspects of a particle's shape: aspect ratio describes the overall shape of a particle. In contrast, convexity is primarily affected by a particle's surface irregularity whereas Cox circularity is affected by both.

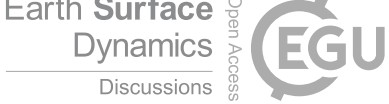



Grain size-shape distributions (SSDs) are constructed from grain size (D2d, Table 2) and the three shape variables, resulting in the distributions named ArD2d, ConD2d and CcD2d. The SSDs are created by assigning individual particles to their respective size-shape classes (Fig. 1C; Table 2). Next, the volume of the grains in each size-shape class is summed, and the distribution is normalised to a sum of 100% using the total volume. This procedure gives rise to three-dimensional distributions

(X = size, Y = shape, Z = volume) (Fig. 1D) that can be visualised as a combination of a grain size (X –Z) and a grain shape (Y-Z) distribution (Fig. 1E).

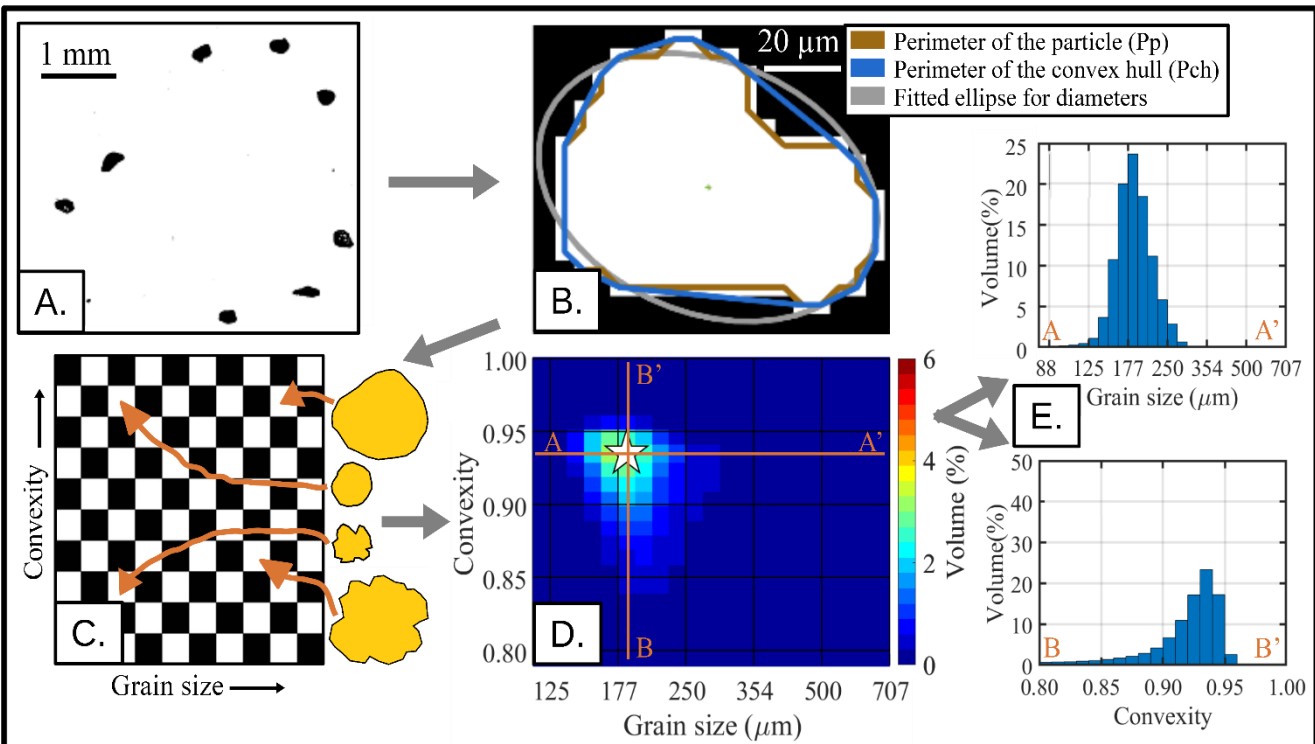

Fig. 1. A: Binary image of sediment grains. B: Computation of particle characteristics (note inversed black-white scale). C: Assignment of grains to size-shape classes. D: Grain size-shape distribution (ConD2d in the example, star marker designates

the mode of the distribution). E: grain size (upper panel) and grain shape (lower panel) cross-sections through the SSD along respectively A-A' and B-B'.

End-member modelling algorithm AnalySize (Paterson and Heslop, 2015) is used to unmix the SSD datasets because it produces the most accurate results of the algorithms currently available (Van Hateren et al., 2017). The computed end members

are hereafter referred to as end member EMx-y where x denotes the end member number from coarse to fine and y denotes the total number of end members in the given end-member modelling solution. For example, the coarsest EM of a dataset with four EMs is referred to as EM1-4.





The fit of end-member modelling solutions to the data is used to infer the most likely number of end members. The fit is described by variance squared, also termed the coefficient of determination ($R^2$). We define two types of $R^2$: 1). class-wise $R^2$, denoting the fit per grain size class (grain-size distributions) or grain size-shape class (SSDs), and 2) sample-wise $R^2$, denoting the fit per sample (Van Hateren et al., 2017). By increasing the number of end members, $R^2$ will increase. However, at a certain

point the increase in fit is not due to geologically significant end members but due to fitting of noise. We therefore seek the minimum number of end members sufficient to explain most of the variation in the dataset. In grain-size data analysis, this minimum number of end members is traditionally estimated by a flattening off of the curve of average $R^2$ versus the number of end members, also known as the inflection point (Prins and Weltje, 1999b; Weltje, 1997). However, tests with artificial grain size data have pointed out that this method sometimes yields an incorrect number of end members (Van Hateren et al.,

2017). Rather than taking the average, we therefore use the full distribution of class-wise $R^2$ to obtain more detailed information on the most likely number of end-members (Prins and Weltje, 1999b; Van Hateren et al., 2017). In addition, we use the distribution of sample-wise $R^2$.

### 2.4 Artificial datasets for testing and validation of the method

Artificial datasets with known end members and end-member abundances (Van Hateren et al., 2019) are used to evaluate 1)

the accuracy of unmixing of SSDs under different mixing scenarios and 2) the potential of, and difference between, class-wise and sample-wise $R^2$ for identification of the most likely number of end members in a SSD dataset. The known end members of the artificial datasets are hereafter referred to as input end members IEMx$_{-y}$ similar to the notation for modelled end members.

Following an approach similar to Van Hateren et al. (2017), three datasets are created with increasingly complex mixing scenarios: The least complex dataset, 3EM_nonoise, is created using as IEMs three samples of the dune dataset with markedly different size-shape distributions (Appendix D). Two-hundred sets of three random numbers are generated with a uniform distribution between 0 and 1 using a random number generator. Each set of three numbers is subsequently normalised to sum-to-one. The two-hundred sets represent the contributions of the IEMs to each artificial sample (end-member abundances). By

multiplying each set of three random numbers with the three IEM SSDs, two-hundred artificial samples are generated.

The second dataset, 4EM_noise, is used to test accuracy of the method in the presence of noise and an additional end member. Addition of noise decreases accuracy of unmixing results in grain size distribution datasets (Van Hateren et al., 2017). The IEMS of the 4EM_noise dataset are the same as those of the 3EM_nonoise dataset except for an additional end member that,

in terms of its grain size, is between the coarsest and intermediate IEM of the 3EM_nonoise dataset (Appendix E). Noise is included in the dataset by multiplying the volume in each size-shape class of the artificial samples by a random number with a normal distribution characterised by a mean of 1 and a standard deviation of 0.05.



The third and most complex dataset, 4EM_noise_highmix, is similar to 4EM_noise but has different end-member abundances. This dataset is used to test the accuracy of the output for highly mixed datasets. In such datasets, one or more of the primary components do not form a dominant contribution to any of the samples. Highly mixed data significantly deteriorate accuracy of unmixing (Heslop, 2015; Van Hateren et al., 2017). We use the following mixing scenario: IEM1$_{-4}$ occurs in only 5 samples at abundances between 0.2 and 1 (20 and 100%). IEM2$_{-4}$, the highly mixed end member, occurs in 100 samples at low abundance between 0.05 and 0.2 (5% and 20%). IEM3$_{-4}$ and IEM4$_{-4}$ occur in all two-hundred samples at randomly varying abundance.

Because the number of end members, the end-member abundances and the end-member SSDs are known, the precision of the unmixing procedure can be deduced from 1) the correlation between IEM SSDs and modelled end-member SSDs, 2) the correlation between the input and modelled end-member abundances and 3) the correlation between the input and modelled data expressed as class-wise and sample-wise $R^2$. Furthermore, the applicability can be assessed of class-wise and sample-wise $R^2$ for identification of the most likely number of end members, which is an unknown in real-world datasets.

## 3 Results

### 3.1 End-member-modelling results for the artificial datasets

#### 3.1.1 End-member-modelling results for the 3EM_nonoise dataset

Due to absence of noise in the 3EM_nonoise dataset, explained variance of the end-member modelling outcome reaches 100 percent at three end members. Because model fit cannot be improved further, the AnalySize algorithm aborts at three end members (the algorithm fits a maximum of 10 end members for real-world datasets that naturally include noise). Figure 2 therefore displays class-wise $R^2$ distributions for results with one to three end members (1EM to 3EM solutions). The 'average' SSD of the dataset as well as the modelled end-members are shown as contours to indicate the relevant size-shape classes. The 1EM solution fits the input data poorly while the 2EM output increases model fit significantly but lacks explanatory power in the size-shape region that coincides with the missing third end member (Fig. 2, Appendix D). Using three end members increases goodness of fit of all size-shape classes to an $R^2$ of 1. Appendix G1 shows median sample-wise and class-wise $R^2$ versus the number of end members. Class-wise $R^2$ shows a near linear increase from one to three end members whereas the curve of sample-wise $R^2$ inflects at two end members. In other words, the improvement in sample-wise $R^2$ is significantly higher from 1 to 2 end members than it is from 2 to 3 end members.





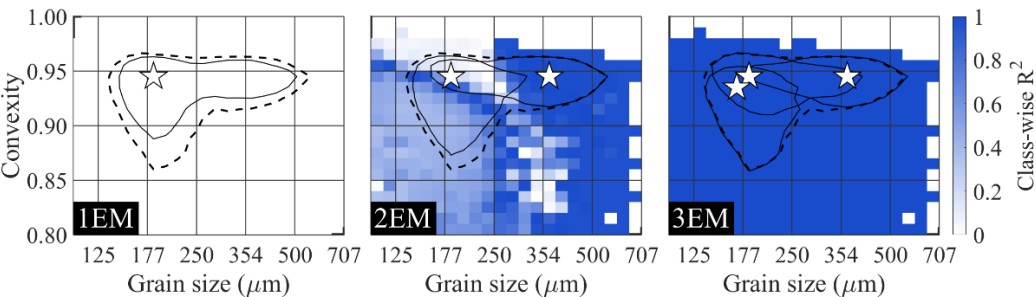

Fig. 2. Class-wise $R^2$ distributions for end-member modelling results of the ConD2d 3EM_nonoise dataset from a 1EM up to a 3EM solution. Solid black lines denote end-member contours: lines drawn along those size-shape classes where the volume of the end-member SSD equals 0.5%. The median SSD of the dataset (average SSD) is represented by a dashed line at 0.2% volume. White stars denote the modes of the end-member size-shape distributions.

Since the 3EM_nonoise dataset is noise-free and consists of three IEMs, an accurate 3EM solution should be identical to the input data, which is nearly the case (Fig. 3). The abundances show 100% explained variance; however, linear trends between the original and determined abundances reveal a slope slightly higher than one, meaning that high input abundances are calculated too high and that low input abundances are calculated too low (below input abundances of approximately three percent, determined abundances go to zero) (Appendix H1). Thus, the computed end members are to a minor degree still mixtures of the IEMs.

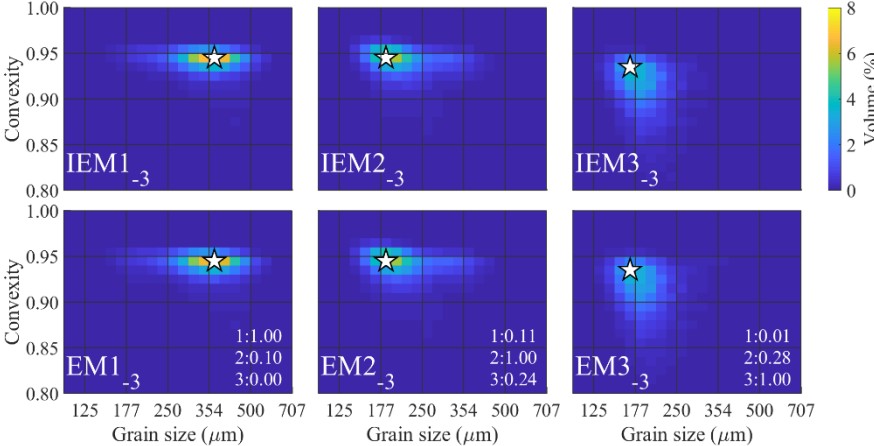

Fig. 3. Input end members (top) and determined end members (bottom) for the ConD2d 3EM_nonoise dataset, with $R^2$ values denoting the fit between them (lower right corners). The first $R^2$ value indicates the correlation of the determined end member with IEM1$_{-3}$, the second with IEM2$_{-3}$ etcetera. White stars mark the modes of the SSDs.





### 3.1.2 End-member modelling results for the 4EM_noise dataset

Figure 4 shows class-wise $R^2$ distributions of solutions for the 4EM_noise dataset. Similar to results for the 3EM_nonoise dataset, a 1EM solution fits the data poorly and a 2EM solution increases the fit significantly but lacks explanatory power in the intermediate and coarse size-shape regions. A 3EM solution fits the intermediate region significantly better but still lacks

explanatory power in the coarse region. Compared to that of the 3EM solution, the class-wise $R^2$ distribution of the 4EM solution displays an increase in $R^2$ in the coarse range because EM1$_{-4}$ more closely resembles IEM1$_{-4}$ than does EM1$_{-3}$ (Fig. 4; Appendix E). The increase in median class-wise $R^2$ is small because the improvement occurs in relatively few size-shape classes (Appendix G2). Median sample-wise $R^2$ similarly increases by a low amount. The increase in sample-wise $R^2$ diminishes from four end members onwards (Appendix G2B). Class-wise $R^2$ even displays a minor decrease of fit. In contrast

to results for the 3EM_nonoise dataset, the explained variance does not reach 100%.

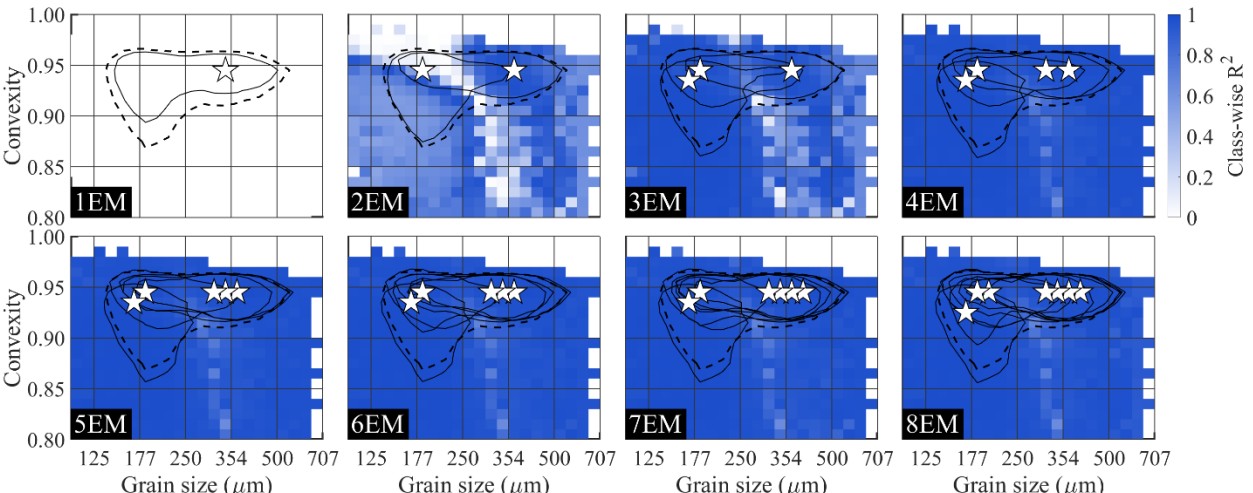

Fig. 4. Class-wise $R^2$ distributions for end-member modelling results of the ConD2d 4EM_noise dataset. Solutions up to eight end members are shown. Solid black lines denote end-member contours: lines drawn along those size-shape classes where the

volume of the end-member SSD equals 0.5%. The median SSD of the dataset (average SSD) is represented by a dashed line at 0.2% volume. White stars denote the modes of the end-member size-shape distributions.

Figure 5 compares the input and determined end-member SSDs. In spite of the noise added to this dataset, the determined end members are very similar to the IEMs. Calculated abundances fit the input abundances well, although a minor scattering is

present (Appendix H2). Similar to the results for the noise-free dataset, linear trends have a slope higher than one, indicating that the determined end members are, to a minor extent, mixtures of the IEMs.

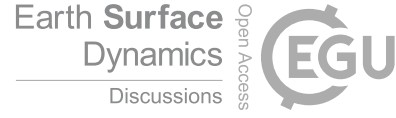

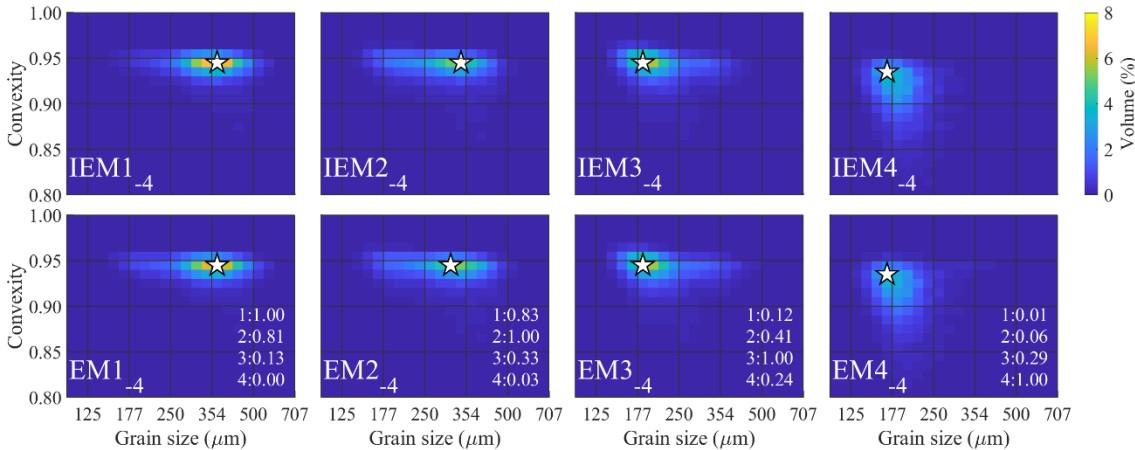

Fig. 5. End members determined for the ConD2d 4EM_noise dataset. Input end members (top) are compared to determined end members (bottom) including $R^2$ values. The first $R^2$ value indicates the correlation of the end member with IEM1$_{-4}$, the second with IEM2$_{-4}$ etcetera. White stars mark the modes of the SSDs.

### 3.1.3 End-member modelling results for the 4EM_noise_highmix dataset

Similar to the results in sect. 3.1.1 and 3.1.2, the 1EM solution modelled for the 4EM_noise_highmix dataset fits the dataset poorly (Fig. 6). The 2EM class-wise $R^2$ distribution is notably different from that of the 4EM_noise dataset: the entire coarse range (>350 μm) is not well reproduced. The reason for this disparity is that IEM1$_{-4}$ and IEM2$_{-4}$ are not represented in this solution and thus the coarse range is underrepresented (Appendix F; Fig. 7).

A 3EM solution covers the coarser range, invoking a strong increase in class-wise $R^2$ to a level comparable to that of the 4EM solution of the 4EM_noise dataset (Fig. 6; Appendix G2 and G3). In contrast to the results for the 4EM_noise dataset, addition of a fourth end member does not result in a significant improvement of class-wise and sample-wise $R^2$.





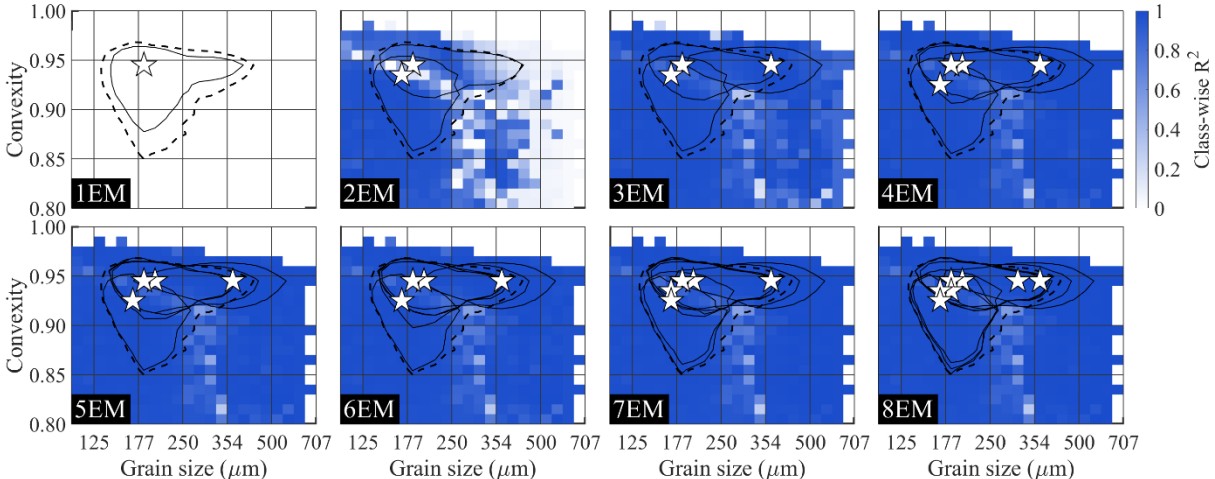

Fig. 6. Class-wise $R^2$ distributions for end-member modelling results of the ConD2d 4EM_noise_highmix dataset. All solutions up to eight end members are shown. Solid black lines denote end-member contours: lines drawn along those size-shape classes where the volume of the end-member SSD equals 0.5%. The median SSD of the dataset (average SSD) is represented by a

dashed line at 0.2% volume. White stars denote the modes of the end-member size-shape distributions.

Size-shape distributions of the end members and IEMs are shown in Fig. 7. The 4EM solution computed for the 4EM_noise_highmix dataset differs in one notable aspect from that calculated for the 4EM_noise dataset: IEM2-4 is not identified as a primary component of the dataset. Rather, the SSD of EM2-4 more closely resembles IEM3-4 leading to

overestimated abundances of EM2-4 and underestimated abundances of EM3-4 in the 4EM solution (Appendix H3). However, the SSDs and the relative abundances of the 3EM solution show a good fit to the SSDs and abundances of the three non-highly mixed IEMs (Appendix F; Appendix H4).

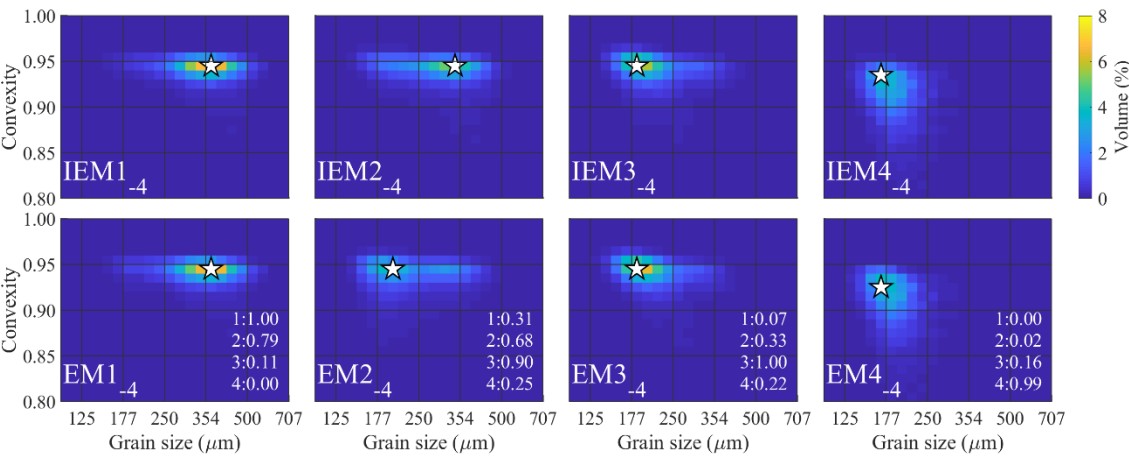

Fig. 7. End members determined for the ConD2d 4EM_noise_highmix dataset. Input end members (top) are compared to

Earth **Surface**
**Dynamics** Open Access
Discussions
EGU

determined end members (bottom) including $R^2$ values. The first $R^2$ value indicates the correlation of the determined end member with $IEM1_{-4}$, the second with $IEM2_{-4}$, etcetera. White stars mark the modes of the SSDs.

## 3.2 Results for the dune dataset

End-member-modelling results for the dune dataset are presented in three subsections: statistics for the ConD2d dataset are shown first to derive the number of end members necessary to explain size-shape variability of the dataset. End-member SSDs and abundances of the robust solution are presented in the second subsection. The third subsection compares results of unmixing based on SSDs to results of unmixing based on grain size distributions (D2d).

### 3.2.1 Unmixing of the size-shape distributions

Appendix I displays the trend of median class-wise and sample-wise $R^2$ against the number of end members. Class-wise $R^2$ reaches a plateau at three end members whereas sample-wise $R^2$ inflects gradually between two and four end members. This gives a first indication that the likely number of end members is between two and four.

Two methods are employed to visualise the fit of the end-member solutions to the dataset in more detail. Class-wise $R^2$
distributions show the fit per size-shape class (fig. 8). The spatial distribution of sample-wise $R^2$ is shown by plotting it on top of an aerial photograph of the study area (Appendix J1). The goodness of fit of the samples is compared to the subregions based on geomorphology as shown in Appendix A3, shown in simplified form in appendix J1 and described in Sect. 2.1. If the unmixing result fits poorly to samples from a specific subregion it is likely that an additional end member is needed to explain the data in that region.

One end member is insufficient to capture the size shape variability of the dataset: the class-wise $R^2$ distribution shows low values across all size-shape classes (Fig.8). The 1EM solution does not fit well to the samples either, as expressed by low sample-wise $R^2$ (Appendix I). The spatial distribution of sample-wise $R^2$ for the 2EM solution shows a good fit to the sediment trap samples of the hinterland. However, the fit to samples of the notch and foredune ridge is poor. The 3EM solution drastically
improves fit in these subregions (Appendix J1). Regarding the class-wise $R^2$ distribution, the 2EM solution performs poorly in the range where its $EM1_{-2}$ and $EM2_{-2}$ overlap, indicating that an additional end member is required to fit these classes (Fig. 8). A 3EM solution represents this intermediate size-shape range much better. Furthermore, comparison of the class-wise $R^2$ distribution to the median data contour shows that this unmixing result performs well in the entire size-shape range where significant volume is present in the data (Fig. 8).

Although the 3EM solution displays high and evenly spread sample-wise $R^2$, there are two regions that stand out: first, slightly lower explained variance occurs at those inland samples that are positioned downwind of fossil dunes that had their vegetation



Earth **Surface**
**Dynamics**
Discussions
EGU

cover removed (Appendix J1; Appendix A2 and A3). A 4EM solution does not improve explained variance of these samples significantly (Appendix J1). Second, the 3EM solution displays low sample-wise $R^2$ on the northern foredune sampling transect, in a small region near the crest (Appendix J1). This is improved by component EM2$_{-4}$ of the 4EM solution which occurs specifically in this region (Appendix L). The specific geographical location of the component indicates that it has some

geological significance. Furthermore, it is also determined in the 5EM and 6EM solutions (Appendix K) and therefore is a robust component. However, it is of minor importance in terms of geographical extent and in terms of the number of samples it represents. The class-wise $R^2$ distribution of the 4EM solution shows amelioration of fit below a convexity of 0.9 and above a size of 250 µm, but volume in this range is insignificant (Fig. 8). Further increasing the number of end members does not increase model fit significantly except that the 6EM solution increases sample-wise $R^2$ for the inland samples downwind of

unvegetated dunes (Appendix J1; Appendix A2 and A3). In conclusion, a 3EM output appears most robust and it reproduces the bulk of spatial variability in grain size and shape, although a four end-member solution locally improves sample-wise $R^2$.

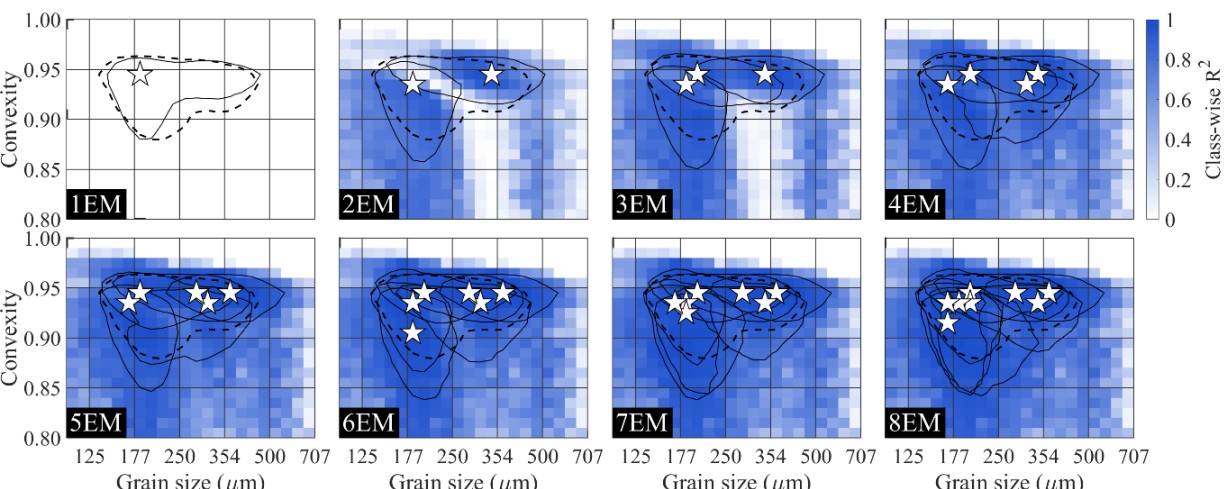

Fig. 8. Class-wise $R^2$ for end-member modelling results of the ConD2d distributions of the dune dataset. All solutions up to

eight end members are shown. Solid black lines denote end-member contours: lines drawn along those size-shape classes where the volume of the end-member SSD equals 0.5%. The median SSD of the dataset (average SSD) is represented by a dashed line at 0.2% volume. White stars denote the modes of the end-member size-shape distributions.

### 3.2.2 End-member composition and abundances of the three-end-member solution

The end-member SSDs of the 3EM solution computed for the ConD2d distribution dataset differ markedly from one another

(Fig. 9): most volume of coarse-grained EM1$_{-3}$ is contained between 250 and 500 µm. Its mode lies at a grain size of 339 µm and a convexity of 0.945. This convexity dominates over the entire size range. The intermediate EM2$_{-3}$ is finer grained, with most of the volume between 160 and 350 µm. Its mode is positioned at a size of 201 µm and a convexity of 0.945. In contrast to EM1$_{-3}$, it shows a gradual decline in convexity with increasing size. Most volume of fine-grained EM3$_{-3}$ lies between 150

and 250 µm. Its mode is located at a size of 185 µm and a convexity of 0.935. It shows a strong decrease in convexity with increasing size.

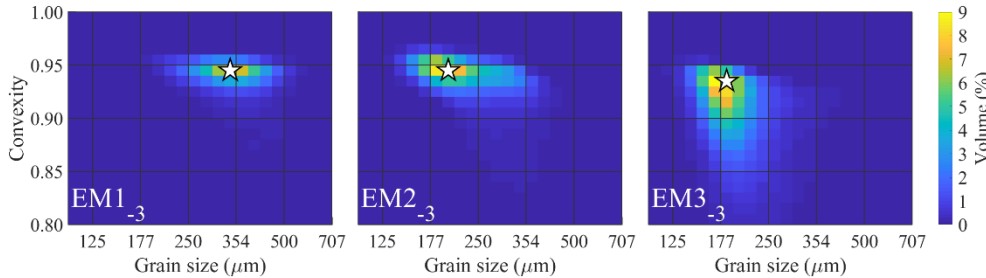

5    Fig. 9. SSDs of the ConD2d 3EM solution determined for the dune dataset. White stars mark the modes of the SSDs.

The end-member abundances of the 3EM solution show a strong spatial differentiation that corresponds with morphological features: EM1$_{-3}$ dominates the unvegetated notch that was dug through the foredune (average abundance 81%). EM2$_{-3}$ dominates most of the sparsely vegetated foredune (average abundance 46%) as well as the vegetated area directly downwind

10    of the sand lobe that is prograding from the notch (average abundance 80%). EM3$_{-3}$ dominates the vegetated hinterland (trap rows B to D, average abundance 94%) (Fig. 10). It is also noteworthy that samples from traps A1, A2 and B2 contain significantly more of EM3$_{-3}$ than the surface samples taken at the same locations and thus also lower the average abundance of EM2$_{-3}$ for the foredune (Fig. 10; Appendix A2).



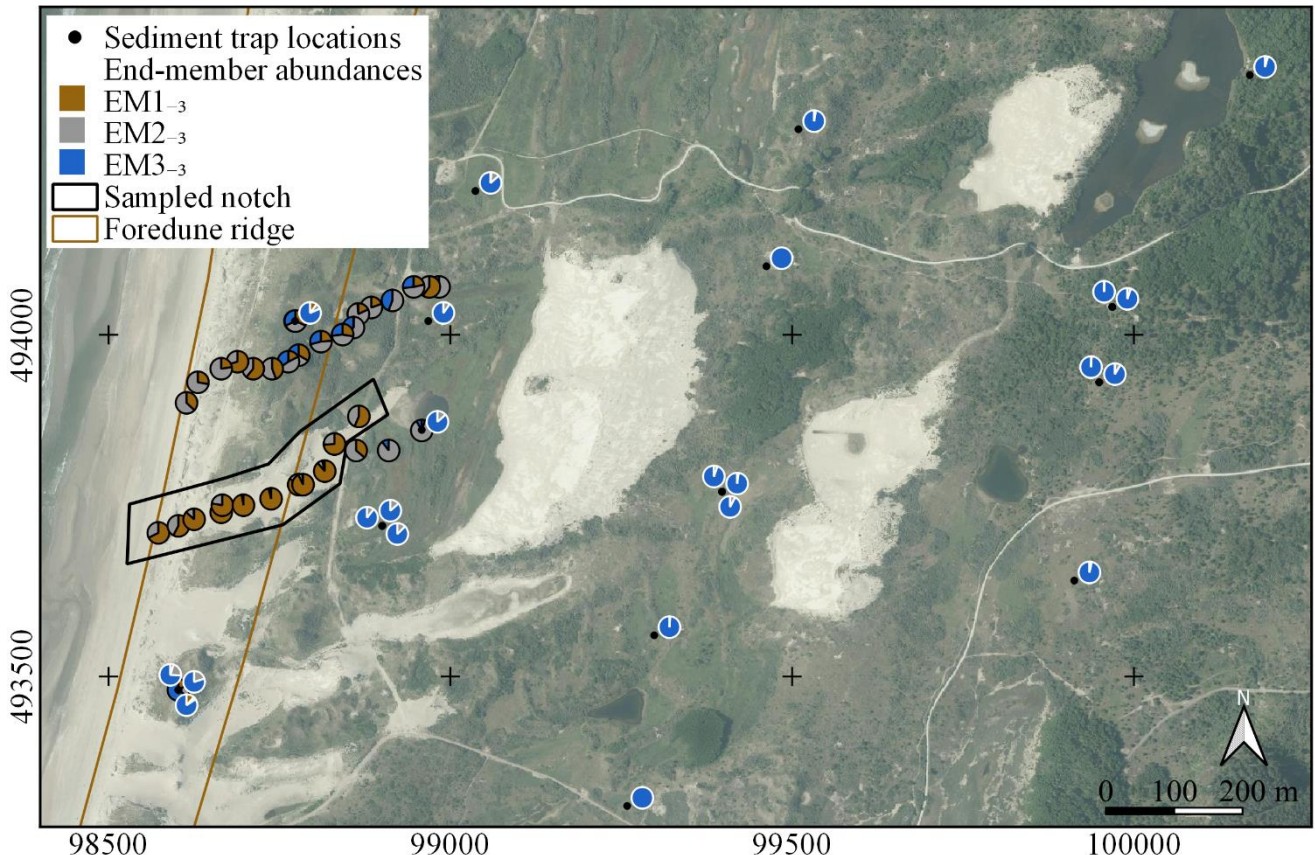

Fig. 10. Pie-charts showing abundances of the 3EM solution computed for the ConD2d distributions of the dune dataset. Pie-charts with a black outline denote surface samples and are plotted at the sampling location. Pie-charts with a white outline denote sediment trap samples and are plotted near the sampling location. The exact locations of sediment traps are marked by black dots. Two of the subregions defined in appendix A3 are shown in simplified form. Aerial photograph © PDOK.nl, 2017.

### 3.2.3 Comparison of results to traditional end-member modelling on grain-size distributions

Besides the size-shape variable ConD2d, we also tested CcD2d and ArD2d. These variables make use of the shape variable Cox circularity and aspect ratio, respectively. In this section we intercompare end-member modelling results of the three size-shape variables. Furthermore, we compare the results using size-shape variables to results from traditional end-member modelling on grain-size distributions (D2d). To enable direct comparison between grain size distributions and SSDs, the latter are transformed to grain size distributions by summation of the volumes of all shape classes per size class and subsequent re-normalisation to 100% (Fig. 11). The 3EM solution is used for the comparison. This number of end members is also robust for traditional grain-size based end-member modelling: median $R^2$ values level off at three end members (Appendix M1), grain



size classes with significant volume show high $R^2$ indicating that class-to-class variability is well resolved (Appendix M2) and sample-wise $R^2$ is high throughout the fieldwork area indicating that spatial variability is also well resolved (Appendix J2).

ConD2d end-member grain-size distributions show significant deviations from those determined for D2d: most notably a finer

modal size for EM2$_{-3}$, but also a more extended fine tail for EM1$_{-3}$ and coarse tail for EM3$_{-3}$ and (Fig. 11A). The grain size distributions of CcD2d show deviations at the same grain-size ranges. However, the deviations are weaker than for ConD2d (Fig. 11B). In contrast, size distributions of the ArD2d end members equal those of D2d (Fig. 11C). Furthermore, the SSDs of ArD2d end members lack the trend in grain shape with grain size that was observed for ConD2d and CcD2d (Fig.9; Appendix N).

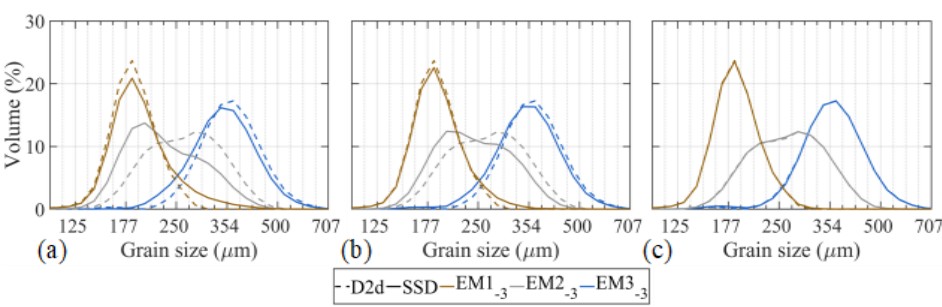

Fig. 11. ConD2d (A), CcD2d (B) and ArD2d (C) 3EM solutions determined for the dune dataset. The SSDs are displayed as grain size distributions (solid lines) and compared to grain size distributions of the D2d 3EM solution (dashed lines).

Table 3 and Appendix O compare end-member abundances for 3EM solutions of ConD2d, D2d, CcD2d and ArD2d. The main trends of all variables correspond: EM1$_{-3}$ prevails in the notch, EM2$_{-3}$ on the foredune and in the vegetated area within 100 m downwind of the notch, and EM3$_{-3}$ in the hinterland. However, differences exist between the variables: ConD2d and CcD2d show higher proportions of EM1$_{-3}$ in the notch than do ArD2d and D2d (Table 3). The four variables show similar proportions of EM2$_{-3}$ on the foredune, but differences occur in the samples directly downwind of the notch. Here, proportions are highest

for ConD2d, followed by CcD2d, D2d and ArD2d (Table 3). Similarly, proportions of EM3$_{-3}$ in the hinterland are slightly higher for ConD2d, followed by CcD2d, ArD2d and D2d (Table 3). In summary, unmixing outcomes of ConD2d are generally most extreme, followed by CcD2d (they show the highest abundances of the dominant end-member). Results from ArD2d and D2d are generally less extreme. This clustering of results agrees with what was observed for the end-member grain-size distributions in Fig. 11: ArD2d distributions are highly similar to those of D2d, whereas ConD2d and CcD2d distributions

differ respectively strongly and weakly from the D2d distributions.

Table 3. Average end-member abundances of the dominant end-member per subregion as defined in Appendix A3.

| Area, prevalent end member | ConD2d (%) | CcD2d (%) | ArD2d (%) | D2d (%) |
|---|---|---|---|---|
| Notch, $EM1_{-3}$ | 81 | 75 | 62 | 62 |
| Foredune, $EM2_{-3}$ | 46 | 47 | 52 | 53 |
| <100m downwind from notch, $EM2_{-3}$ | 80 | 66 | 57 | 58 |
| Hinterland, $EM3_{-3}$ | 94 | 90 | 90 | 89 |

## 4 Discussion

### 4.1 Accuracy of end-member modelling on size-shape distributions

#### 4.1.1 Accuracy of the unmixing methodology under different mixing scenarios

The precise 3EM solution for the 3EM_nonoise dataset confirms that the method is highly accurate under the condition that no noise is present in the dataset. Results for the 4EM_noise dataset indicate that computed end members remain correct reproductions of the input end members in presence of noise. However, the noise induces minor deviations in the end-member proportions. Two conclusions can be drawn on basis of the results for the 4EM_noise_highmix dataset. First, primary components that occur in a limited number of samples but at high proportions ($IEM1_{-4}$) can be accurately determined by AnalySize. Second, highly mixed primary components ($IEM2_{-4}$) cannot be determined accurately by AnalySize. This outcome is similar to results for highly mixed grain-size distribution data (Van Hateren et al. 2017). The implication for real-world datasets is that highly mixed components will be overlooked during the end-member modelling procedure. However, our results indicate that the remaining end members and their relative proportions are computed accurately.

#### 4.1.2 Methods for determination of the most likely number of end members

In the current study we use artificial datasets with a known number of end members. This allows us to test three methods for detection of the statistically feasible number of end members: median class-wise $R^2$, median sample-wise $R^2$ and class-wise $R^2$ versus size and shape (a class-wise $R^2$ distribution). The latter is similar to a graph of class-wise $R^2$ versus grain size for grain size data.

Our results for artificial datasets indicate that interpretation of the number of end members is straightforward in the absence of noise but ambiguous when noise is present: the noise-free dataset (3EM_nonoise) displays class and sample-wise $R^2$ values of one when the number of determined end members equals the number of end members present in the dataset. In contrast, the $R^2$ values for the noise-containing dataset (4EM_noise) never reach 1, which is more in line with end-member modelling





results for real-world datasets. In this case, median $R^2$ can only be used as a rough indication of the number of end members since an 'inflection point' (Prins and Weltje, 1999b; Weltje, 1997) is ill-defined: median $R^2$ values for the dataset level off at three end members rather than four. A class-wise $R^2$ distribution provides a better estimation of the number of end members: the presence of four end members is apparent from an increase in class-wise $R^2$ in the coarser size range going from a 3EM to

a 4EM solution. The presence of the highly mixed end member in the dataset 4EM_noise_highmix is not apparent from the class-wise $R^2$ distribution, indicating that such an end member will likely be ignored in the end-member modelling of real-world data.

There are two additional conceivable methods for determination of the geologically feasible number of end members: 1) A

graph of sample-wise $R^2$ against depth (core/outcrop) or against sample location (spatial data such as the dune dataset) and 2) using samples of known origin to demonstrate the geological meaning of the end members (Weltje and Prins, 2003). These two methods cannot be tested with artificial data and thus will be discussed using the dune dataset.

Results for the dune dataset indicate that spatially resolved sample-wise $R^2$ can be used to determine the number of end

members, especially when the spatial distribution of model fit is compared to known geomorphology of the area. For example, the 2EM solution fits poorly to the samples of the notch and foredune area. This indicates that two primary components are insufficient to describe the processes occurring in these subregions. The 3EM solution satisfactorily fits all main subregions, indicating that it captures the main transport processes that are active in the study area. The dune dataset also provides two examples of modern-day samples of known transport processes that can be used as reference material for paleo-studies. Surface

samples from the notch area can be used as a reference for aeolian bedload sediment because the surface of the notch area was characterised by aeolian current ripples. Furthermore, samples from sediment traps, especially from rows C and D which are furthest land-inward (appendix A), can be used as a reference for aeolian suspension because 1) the distance from the main source areas (beach/ notches) excludes modified saltation from reaching the traps, 2) land-inward from the foredune ridge, denser vegetation rules out new entrainment of sediment (Arens et al., 2002; Lancaster and Baas, 1998) and 3) the height of

the sediment traps further reduces the chance of contamination by local saltation.

## 4.2 The value of end-member modelling on size-shape distributions: implications of the dune dataset

### 4.2.1 Geological significance of the three-end-member ConD2d model

The spatial distribution of end members of the 3EM solution relates strongly to the geomorphology of the area: EM1$_{-3}$ occurs

mainly on the bare surfaces of the beach and notch, EM2$_{-3}$ occurs on the sparsely vegetated foredune and within the vegetated area directly downwind of the notch, and EM3$_{-3}$ occurs in the vegetated hinterland. This geographical differentiation suggests that the end members are linked to the three aeolian processes known to operate on a beach to dune transect: 1) bedload,



consisting of saltation, reptation and creep, the motions of which are predominantly affected by gravity, 2) modified saltation, which is affected by both gravity and turbulence and 3) suspension, of which the motions are predominantly affected by turbulence (Arens et al., 2002; Hunt and Nalpanis, 1985).

As mentioned in Sect. 4.1.2, aeolian current ripples on the supratidal beach and in the notch confirm that EM1$_{-3}$ is linked to the bedload population. Component EM2$_{-3}$ specifically occurs on the windward and leeward slope of the foredune. Several processes on the foredune increase the proportion of grains travelling in modified saltation (Arens et al., 2002): 1) On the windward slope of the foredune, relief and marram grass induce turbulence, thereby increasing the proportion of grains that travel in modified saltation and suspension. 2) At the same time, the vegetation partly impedes bedload transport. 3) At the

foredune crest, flow separation induces even stronger vertical air motion, forcing the grains into short-term suspension. The grains that are less susceptible to turbulence are deposited at the leeward side of the foredune (modified saltation population), whereas the grains that are more susceptible to turbulence (the true suspension population) travel further land inward where EM3$_{-3}$ dominates. As stated in Sect. 4.1.2, the interpretation of EM3$_{-3}$ as suspension component is further corroborated by the distance from the source (beach/notches), the dense vegetation in the hinterland, and the fact that the sediment traps are at

approximately 1.5 m above ground level. Sediment traps om the foredune also show a high contribution of EM3$_{-3}$, which is on average higher than that of the surface samples at the same location. This is likely related to the height of the traps, causing them to trap the sediment that is in transport (suspended load and modified saltation) rather than the sediment that is deposited (bedload and modified saltation).

The three end members were also set apart by a markedly different *shape* of their size-shape distributions: the bedload population was characterised by a constant grain regularity with increasing size, the modified saltation population by a minor decrease in grain regularity and the suspended population by a strong decrease in grain regularity. These differences are likely caused by differences in size-shape sorting between the transport modes. Movements of grains in saltation are driven mainly by gravity (Hunt and Nalpanis, 1985), which is a function of particle mass. Because the beach sediments in our fieldwork area

are of uniform density with negligible heavy mineral content (Eisma, 1968), particle mass is mainly determined by particle size. Size, not shape, is therefore the predominant sorting agent during saltation. Eisma (1965) furthermore inferred that it is likely that surface creep favours spherical grains because they roll more easily. It therefore follows that the overall bedload population should show relatively regularly shaped grains and no significant trend of grain shape with grain size. This is indeed the case for EM1$_{-3}$.

Settling of grains in suspension is driven by gravity and restrained by aerodynamic drag of a particle. The latter factor also depends on grain shape: irregular grains have more drag, and thus settle slower (Komar and Reimers, 1978) and are also more susceptible to turbulence. It therefore makes sense that the SSD of suspension component EM3$_{-3}$ shows a strong decrease in grain regularity with increasing size: the irregularity of the coarser grains compensates for their larger weight. Chinese loess

Earth **Surface**
**Dynamics**
Discussions

deposits are on the order of two to ten times finer grained than EM3$_{-3}$ and show a similar decrease in grain regularity with increasing size (Shang et al., 2018). This indicates that: 1) a decrease in grain regularity with increasing size is characteristic of sediments transported in aeolian suspension, and 2) for a given transport mode and a similar grain shape range, the grain-size of sediment depends on, and is a reflection of transport conditions (amount of transport energy available and transport distance). SSDs are therefore a good indication of the mode of transport; grain-size distributions are not.

Modified saltation is a process that is intermediate between saltation and suspension: grains are saltating (sorted by susceptibility to gravity) but are also shortly suspended (sorted by susceptibility to gravity and turbulence). The size-shape distribution of EM2$_{-3}$ is indeed intermediate between EM1$_{-3}$ and EM3$_{-3}$, both in terms of its grain size and its minor decline in grain regularity with increasing grain size.

### 4.2.2 A comparison of traditional grain-size based and novel size-shape based end-member modelling

End member distributions obtained using size-shape variable ArD2d are remarkably similar to those obtained using traditional size-based end-member modelling (D2d). This suggests that during transport, grains are not sorted by their aspect ratio. However, Shang et al. (2018) did observe sorting of aspect ratio. This incongruity may be explained by the difference in how aspect ratio was defined in the two studies: We defined aspect ratio based on the major and minor diameters of ellipses fitted to the particles. These diameters represent the overall particle shape since their length is not sensitive to small-scale particle roughness: the ellipse fitting procedure 'averages out' small humps. In contrast, the major and minor Feret diameters as used in Shang et al. (2018) are affected by such small humps.

In contrast to ArD2d, end-member modelling results of CcD2d and especially ConD2d differ from D2d (grain size): the mode of their intermediate end member is significantly finer-grained and it overlaps more substantially with EM3$_{-3}$. This overlap may actually be the cause of the observed difference: end-member modelling on size-shape distributions would be more suitable for identification of an end member that strongly overlaps with another in terms of grain size but differs in grain shape. Of the three studied size-shape variables, results of ConD2d shows the strongest unmixing (highest abundances of the dominant end-member). This indicates that ConD2d may be the most appropriate variable for the identification of transport processes.

### 5 Conclusions

We introduce a novel method that can be used to reconstruct sediment transport processes from sedimentary deposits. The method makes use of end-member modelling on grain size-shape distributions, which are constructed from grain size and shape data obtained by dynamic image analysis. Tests with artificial size-shape distribution datasets indicate that the known

end-members and end-member mixing proportions are accurately computed by the method, even when noise is present in the data. End-members with limited occurrence are also identified; highly mixed components, however, cannot be determined accurately. The tests also point out that the distribution of the fit of unmixing results per size-shape class (the class-wise $R^2$ distribution) can be used to indicate the number of end-members present.

The size-shape distribution unmixing method is also applied to real-world data from an active aeolian system in the Dutch coastal dunes. Results show that a comparison of the spatial distribution of model fit (sample-wise $R^2$) to local geomorphology further increases insight into the number of end-members present. The geological meaning of end members can be validated by comparing their size-shape distributions to reference samples of different transport processes.

Three end members are determined for the dune dataset. The spatial distribution of these end members is in accordance with the local geomorphology and reflects the three dominant aeolian transport processes known to occur along a beach to dune transect: bedload, modified saltation and suspension. These processes are characterised by distinctly different end-member size-shape distributions, resulting from differential (size and) shape sorting: with increasing size, bedload shows a constant

grain shape, modified saltation a minor decrease in grain regularity, and suspension a strong decrease in grain regularity (when using convexity or Cox circularity as shape parameter).

Compared to traditional end-member modelling on grain-size distributions, unmixing of SSDs gives rise to different end-member grain-size distributions due to shape sorting effects. Results of the new method also show higher proportions of the

20 dominant end members, indicating a better discrimination of the aeolian transport processes (especially when using convexity as shape parameter). The principal advantage of the new method, however, is that the characteristic shapes of the end-member size-shape distributions can be used as a fingerprint of the transport mode. The new method therefore resolves the ambiguity that arises when the transport mode is reconstructed using grain-size distributions.

**Code availability**

Not available

**Data availability**

Data are available from the Pangaea database (https://issues.pangaea.de/browse/PDI-21911)



**Appendices**

Appendix A. Fieldwork area in the coastal dunes of the National Park Zuid-Kennemerland. A1 shows the general location of the study area. A2 displays the locations of surface samples and sediment traps. A3 covers the same area and shows subregions based on geomorphic features. Aerial photograph © PDOK.nl, 2017.



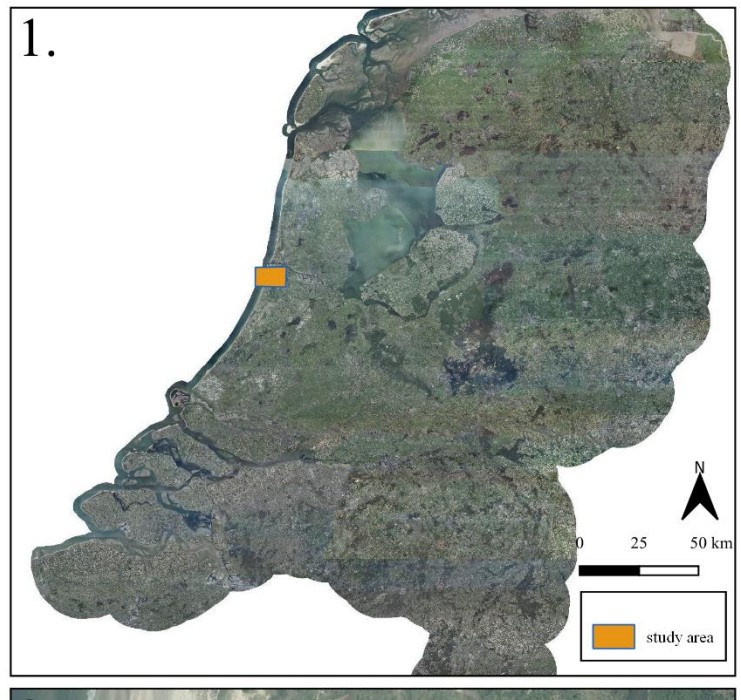

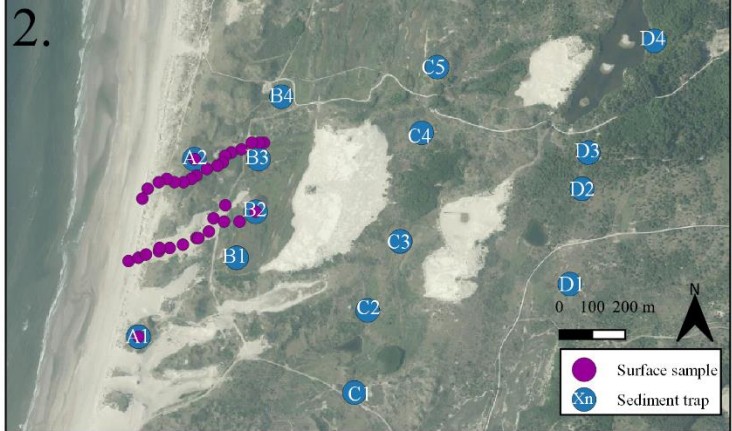

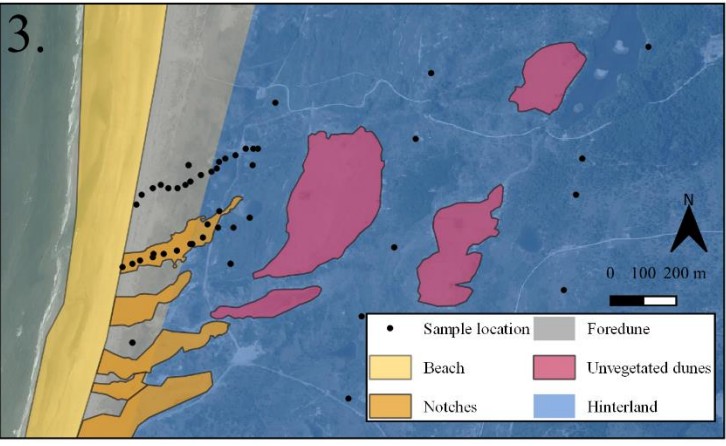



Appendix B. Sediment trap on a vegetated dune

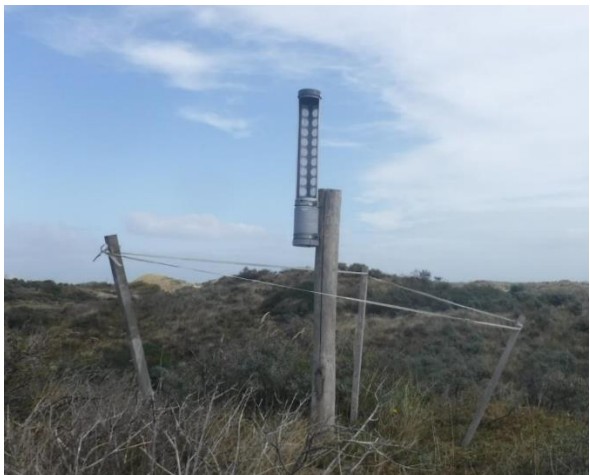



Appendix C. Flow diagram for the image processing script.

**1. Settings**
1.1. Maximum number of frames to analyse ($1.5*10^4$)
1.2. Maximum number of particles to analyse ($1*10^6$)
1.3. Specify pixel size (4.82 μm)
1.4. Specify cut-off size (13 μm)

**2. Loop per sample**
2.1. Import video file

**3. First loop per frame**
3.1. Store frame as binary image (matrix of zeroes and ones)
3.2. Clear all particles connected to the image boundary
3.3. Fill empty pixels within particles by converting to ones all zeroes that are enclosed by ones (Matlab function *imfill*) (necessary for highly transparent quartz grains in the dune sands).
3.4. Save modified frame

**4. Second loop per frame**
4.1. For all particles in the current frame, obtain major axis $D_A$ and minor axis $D_B$. The directions and lengths of these axes equal the eigenvectors and eigenvalues of the covariance matrix of the particle's pixel locations (Matlab function *regionprops*).

4.2. Create lists with X and Y coordinates of the particle's boundary pixel centres.

**4.3. Compute perimeter (loop per particle)**
4.3.1. Compute absolute distances between boundary pixels in X and Y direction using the lists of step 4.2.
4.3.2. Compute absolute distances between boundary pixels (Pythagorean theorem)
4.3.3. Sum the absolute pixel distances to obtain the total perimeter of the particle (Pp).

**4.4. Compute area (loop per particle)**
Compute the area within the boundary pixels using the lists of step 4.2.

**4.5. Compute length of convex hull (loop per particle)**
4.5.1. Obtain all convex points along the boundary of the particle using matlab's function *convhull* on the lists of step 4.2.
4.5.2. From the convex hull points, compute the total length along the convex hull similar to step 4.3.1. to 4.3.3.

**5.** Store all particle data of current frame in a large matrix

**6.** Scale all particle properties computed thus far using the pixel size given in the settings section
**7.** Compute D2d, volume, AR, Con and Cc based on the properties computed thus far (see Table 1 for the equations).
**8.** Delete all particles with size smaller than the cut-off size given in the settings section.
**9.** Save the particle properties matrix to computer.





Appendix D. Input and determined end-member SSDs for the 3EM_nonoise dataset.

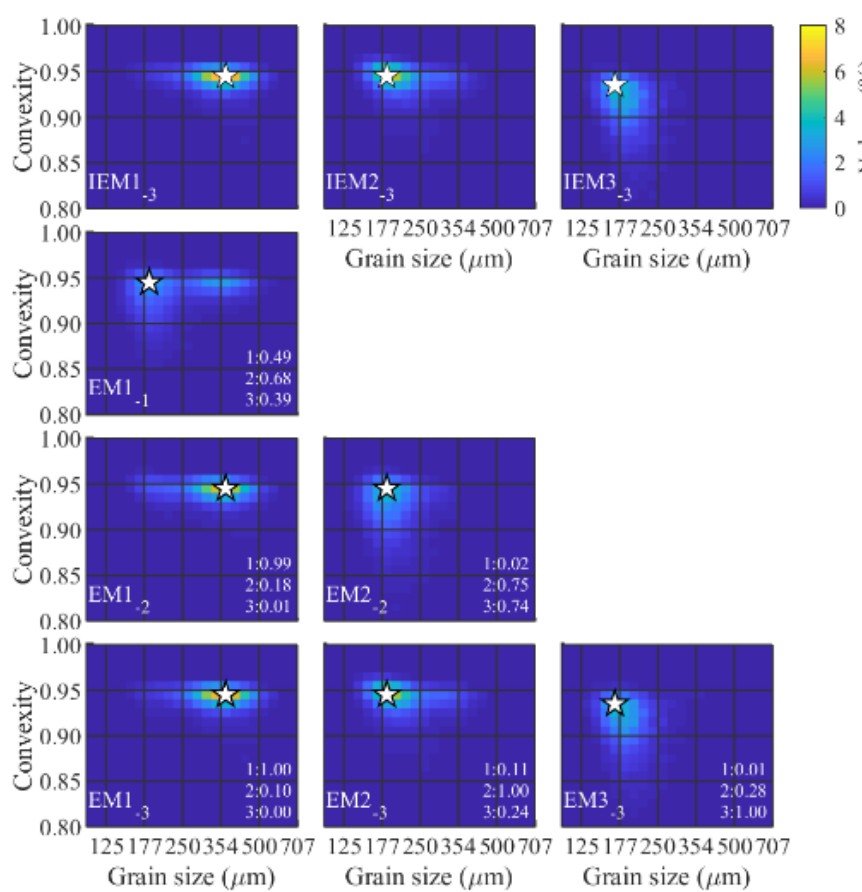





Appendix E. Input and determined end-member SSDs for the 4EM_noise dataset.

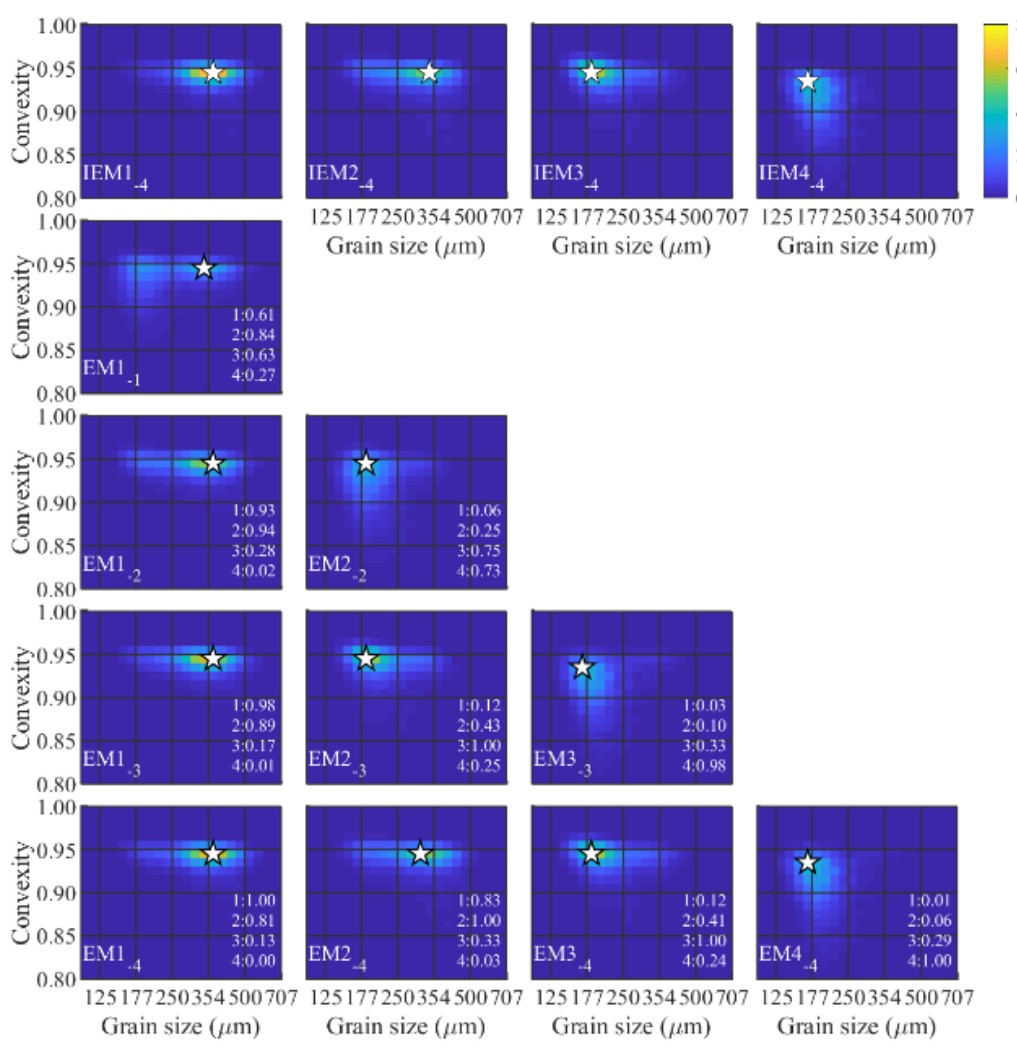





Appendix F. Input and determined end-member SSDs for the 4EM_noise_highmix dataset.

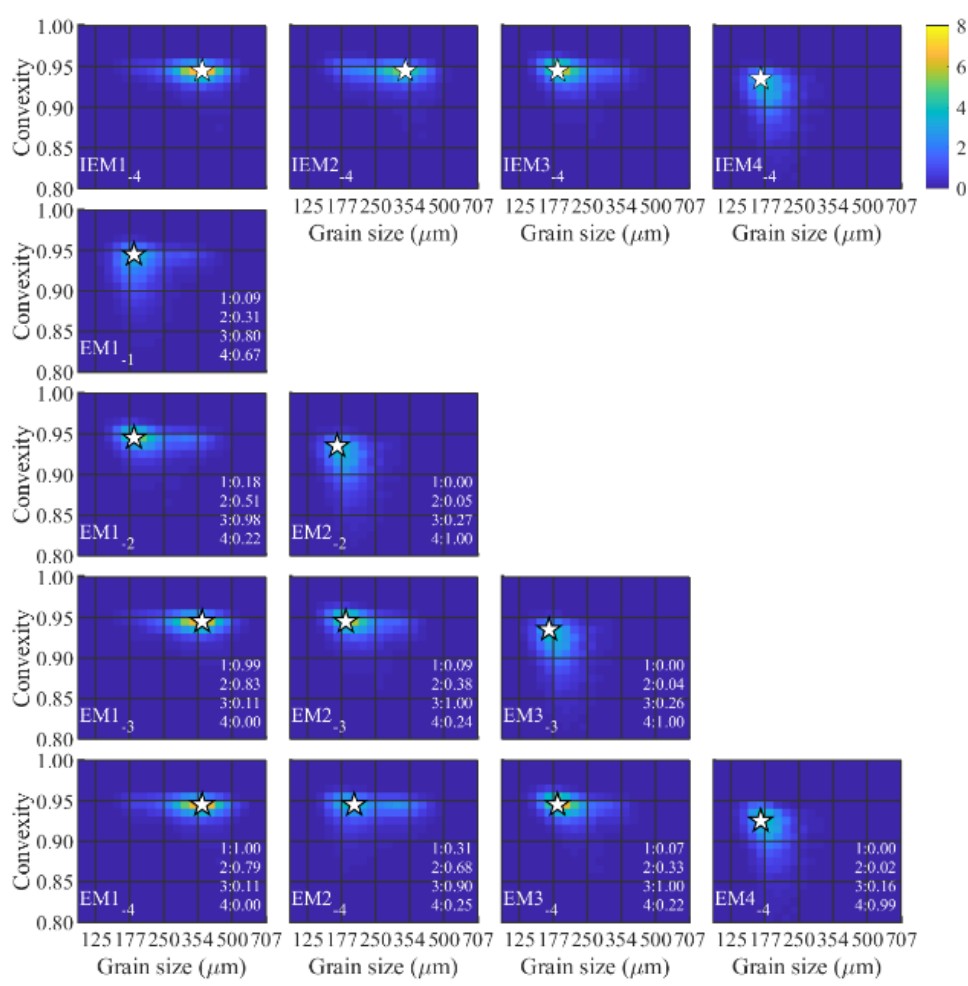





Appendix G. Median class- and sample-wise $R^2$ versus the number of end members for the artificial datasets: 3EM_nonoise (1), 4EM_noise (2) and 4EM_noise_highmix (3). Figure 2B and 3B zoom in on Fig. 2A and 3A.

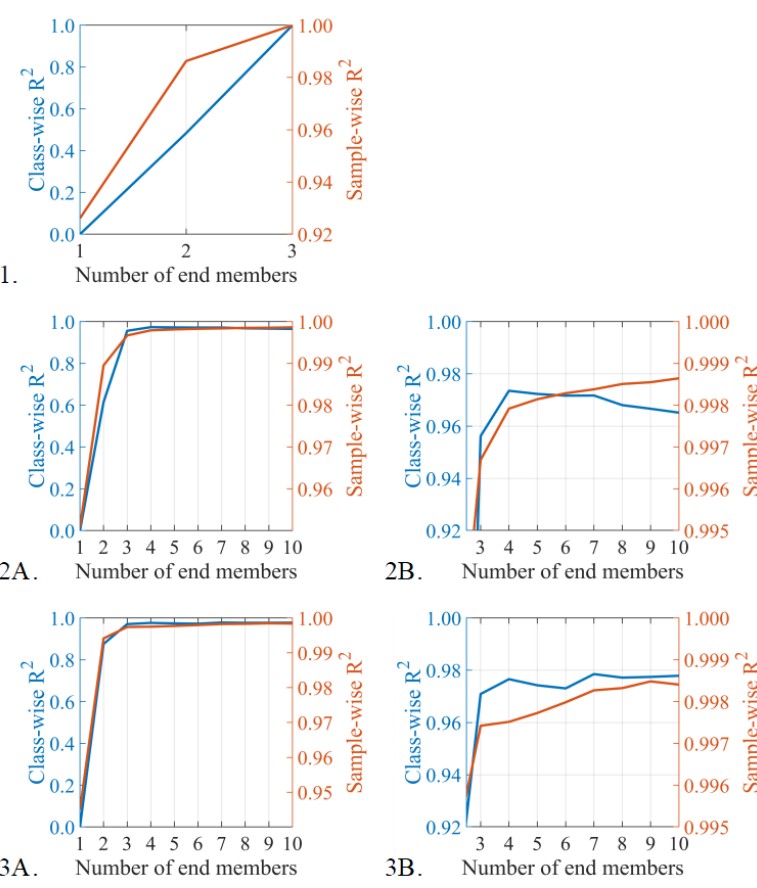



Appendix H. A comparison of modelled and input end-member abundance for the artificial datasets (1: 3EM solution for 3EM_nonoise, 2: 4EM solution for 4EM_noise, 3: 4EM solution for 4EM_noise_highmix, 4: 4EM solution for 4EM_noise_highmix but without the highly mixed end-member).

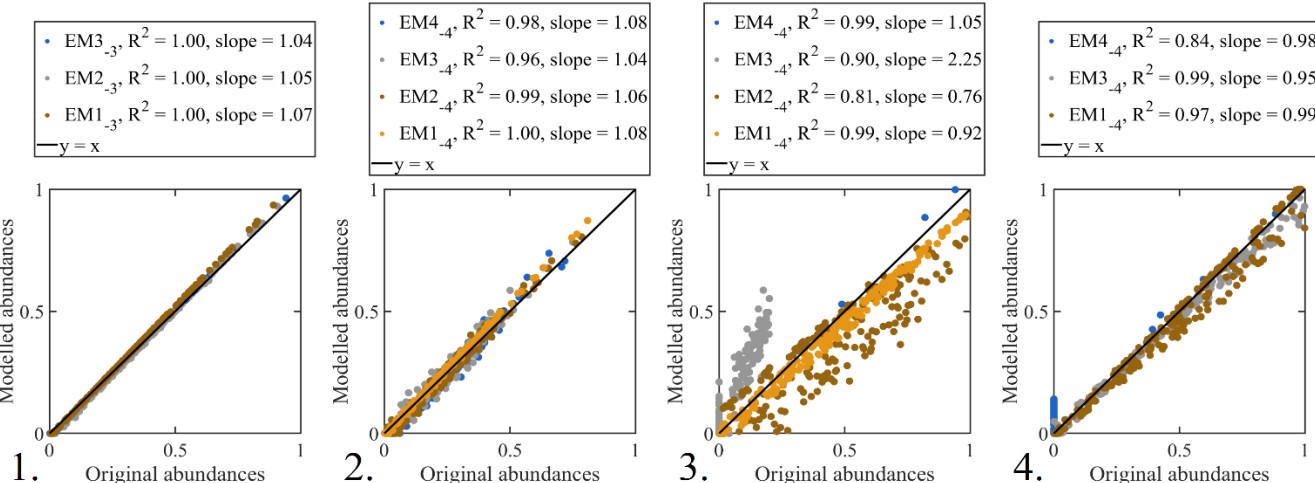

Appendix I. Median class- and sample-wise $R^2$ versus the number of end members for the ConD2d distribution dune dataset.

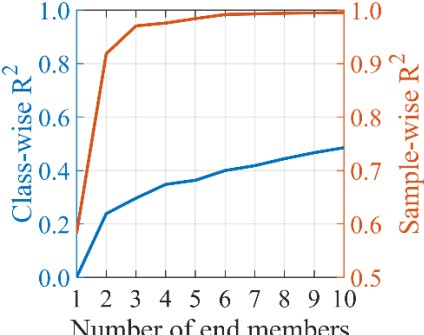

Appendix J. Sample-wise $R^2$ plotted over the sample locations for variable ConD2d (J1) and variable D2d (J2). The first interval of the colour scale is enlarged to elucidate the changes in $R^2$, which mainly occur above a value of 0.9. Points with a black outline denote surface samples and are plotted at the sampling location. Points with a white outline denote sediment trap samples and are plotted near the sampling location. The exact locations of sediment traps are marked by white dots. Two of the subregions defined in appendix A3 are shown in simplified form. Aerial photograph © PDOK.nl, 2017.









Appendix K. End-member distributions computed for the ConD2d dune dataset (1 to 8EM solutions).





Appendix L. End-member abundances for variable ConD2d determined for the dune dataset. Pie-charts with a black outline denote surface samples and are plotted at the sampling location. Pie-charts with a white outline denote sediment trap samples and are plotted near the sampling location. The exact locations of sediment traps are marked by black dots. Two of the subregions defined in appendix A3 are shown in simplified form. Aerial photograph © PDOK.nl, 2017.

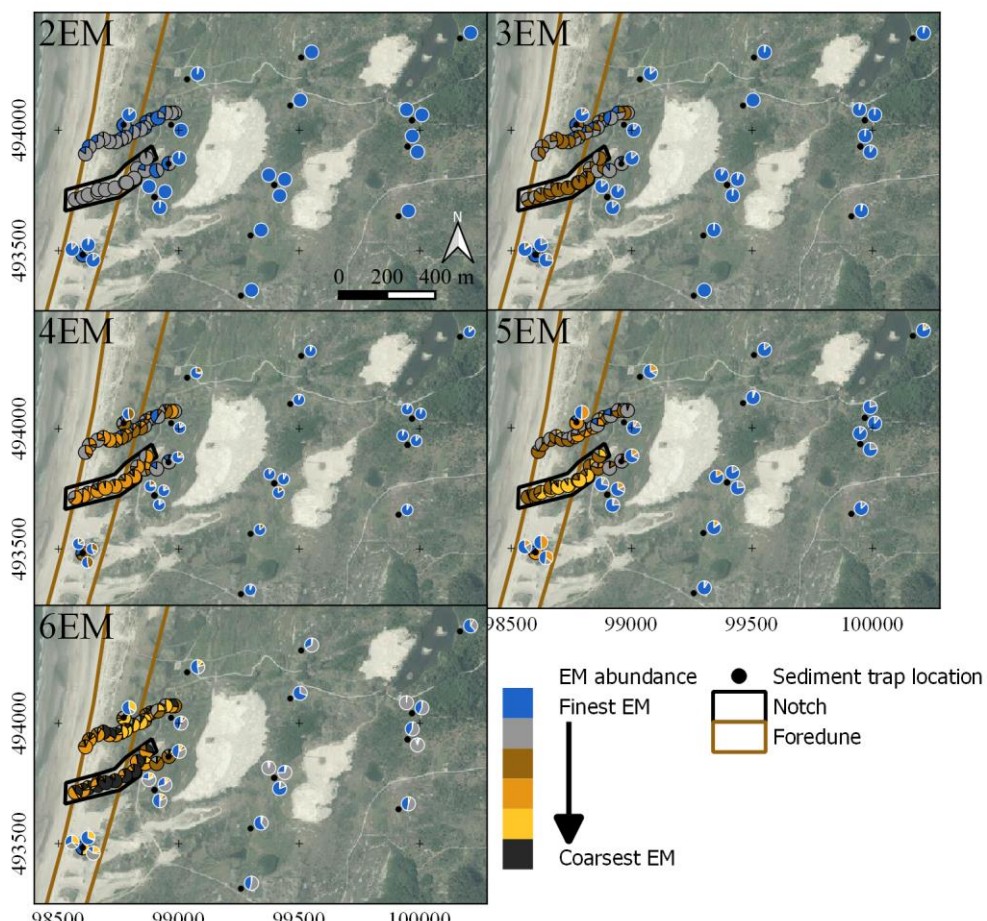

Appendix M. Median class-wise and sample-wise R² for D2d end-member modelling results of the dune dataset (1) and size-resolved class-wise R² (2).

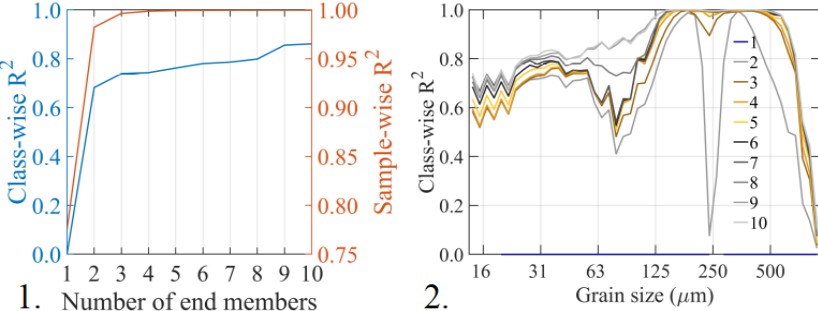



Appendix N. CcD2d (1) and ArD2d (2) 3EM solutions determined for the dune dataset.

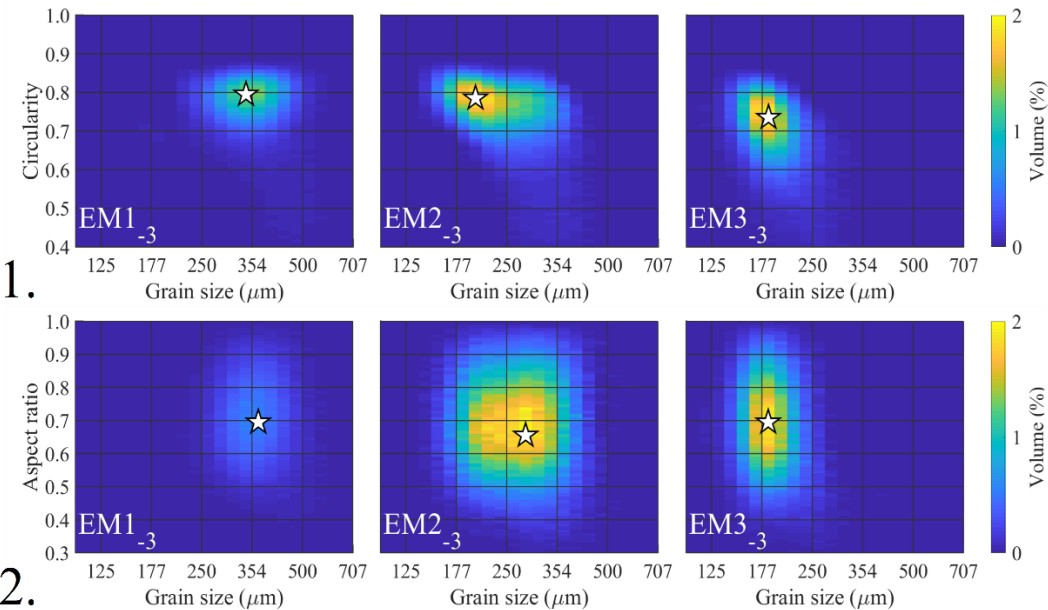

Appendix O. Pie-charts showing abundances of the 3EM solution determined for the ConD2d (1), CcD2d (2), ArD2d (3) and D2d (4) distributions of the dune dataset. Pie-charts with a black outline denote surface samples and are plotted at the sampling location. Pie-charts with a white outline denote sediment trap samples and are plotted near the sampling location. The exact location of sediment traps is marked by black dots. Two of the subregions defined in appendix A are shown in simplified form. Aerial photograph © PDOK.nl, 2017.



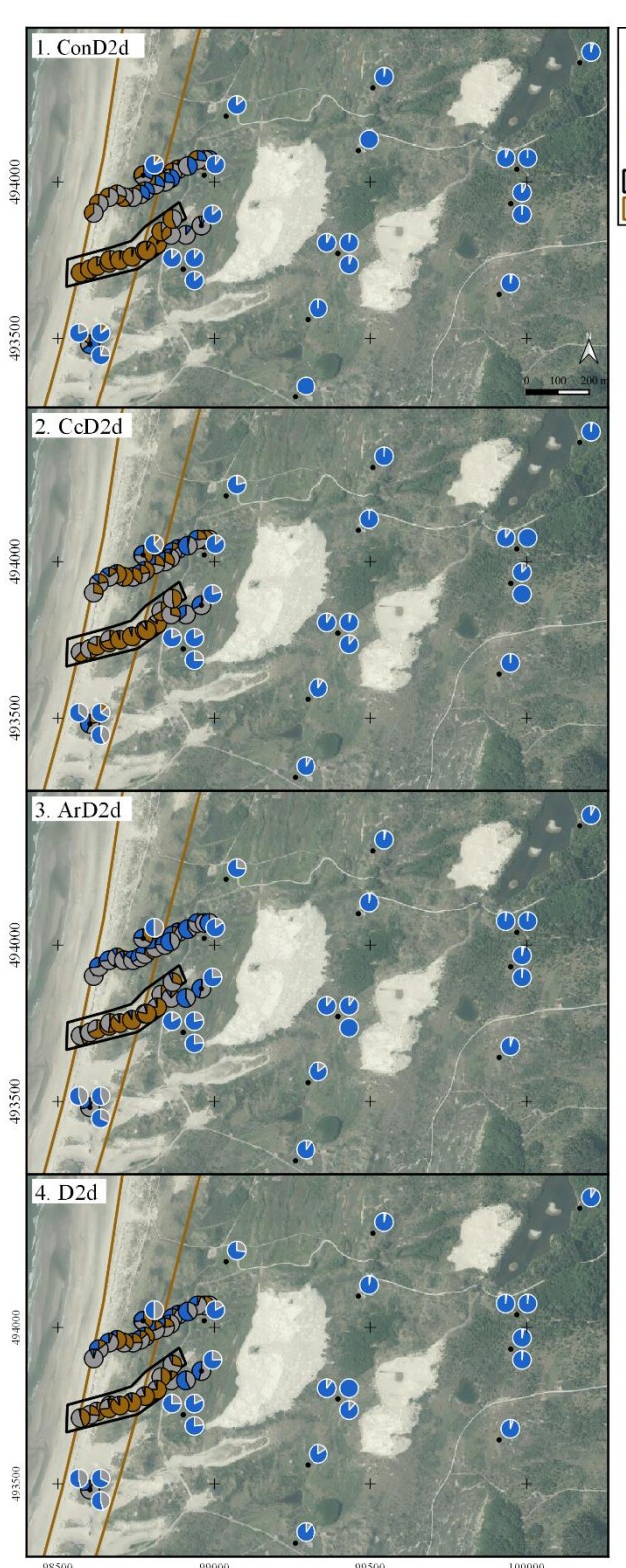



**Author contributions**

JAH wrote the initial draft of the paper, devised a way to integrate grain size and shape data, designed the code used for image processing, data processing and data visualisation, and performed laboratory analyses. UVB performed fieldwork in the National Park Zuid-Kennemerland, laying the foundation for the dune dataset. He also performed laboratory analyses, performed initial tests on shape sorting during aeolian transportation and helped improve the draft versions of the paper. SMA initiated and continued the sediment monitoring project in the National Park Zuid-Kennemerland using sediment traps. He also helped improve the initial draft of the paper. RTVB helped improve writing and construction of the manuscript and provided feedback on the method. MAP conceived the idea of end-member modelling of integrated size and shape data for a better understanding of sedimentological processes, performed fieldwork in the National Park Zuid-Kennemerland, initiated grain size and shape analysis on the dune sediments, helped improve the draft versions of the paper and provided feedback and discussion on how to implement the method.

**Competing interests**

The authors declare that they have no conflict of interest.

**Acknowledgements**

We would like to thank Jeroen van der Lubbe (Research associate in Palaeoceanography/ Palaeoclimatology at Cardiff University and visiting fellow at the Vrije Universiteit Amsterdam) for his contributions to the image processing script and general discussion on image processing of sedimentary grains. We would furthermore like to thank Kay Beets (Vrije Universiteit Amsterdam) for his insights and his role in student projects in the Dutch coastal dunes. Martine Hagen (Vrije Universiteit Amsterdam) is thanked for general support during execution of the labwork. We would also like to express our gratitude to Marieke Kuipers and Hubert Kivit of water company PWN for granting permission to perform the study in the National Park Zuid-Kennemerland, for their encouragement and for their general support. Furthermore, we thank Gerard van Zijl (PWN) for maintenance of the sediment traps and collection of sediment trap samples. Bachelor students Nick van der Veen, Paul Much and Marenqo van der Noll are thanked for fieldwork and sample collection in the National Park Zuid-Kennemerland.

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
