# Peer review of "Identifying sediment transport mechanisms from grain size-shape distributions"

_Earth Surface Dynamics, 2019_

## Referee Comment (RC1) · Anonymous Referee #1 · 8 Jan 2020

General notes

The paper entitled 'Identifying sediment transport mechanisms from grain size-shape distributions' by van Hateren et al. deals with a new methodological approach to determine different transport processes of dune formation in an active aeolian system in The Netherlands. The authors applied particle size and shape data from dynamic image analysis and end-member modelling. The measurements techniques, data handling and data process approaches, as well as the end-member modelling technique are excellent examples from the modern sedimentology and geomorphology. As dynamic image analysis (and automated image analysis, in general)-based approaches

are quite new methods in land surface process investigations and sedimentary studies, a more detailed description of the technique is suggested (Some suggested points: Why are these methods better than previous sizing techniques? What are the advantages and drawbacks compared to other imaging approaches? The repeated pumping of the sample causes data redundancy [one particle will appear on more than one captured frames]. Is it a problem, or not?).

Direct and objective granulometric (size and shape) measurement of large number (n>10ˆ5-10ˆ6) of sedimentary particles is only feasible by using automated image analysis techniques. The applied dynamic image analysis settings provide valuable information on size and shape of randomly oriented (relatively coarse: medium, coarse silt and sand) grains. The description and the presented flow diagram ensure the reproducibility of the introduced measurement and data processing approach for experts of the field, but researchers without deeper knowledge on image analysis-based grain size and shape characterization may have trouble to understand the key steps of the method.

The applied self-written Matlab script (with imfill, regionprops and convhull functions) is using the raw images of the acquired frames not processed data of the Sympatec Qicpic's software, allowing a more detailed and freely customizable data handling. Are there any differences among the results of your own calculations (by using 'regionprops') and ones by the device software?

The presented size-shape distributions are equivalent to volume-weighted scatter plots of size and different shape parameters of individual particles. (It is a relatively well known approach of image analysis-based granulometric characterization, actually, a default data visualization mode in the software of Malvern Morphologi automated static image analyser device).

End-member modelling to separate the different transport processes is applied on the size-shape distributions. The appropriateness of this unique (and great) approach

is also demonstrated by unmixing of three artificial datasets results of increasingly complex mixing scenarios (3-4 EMs with/without added noise). The results are presented clearly and are expressing undoubtedly the correctness of the newly introduced methodology. Limitations of the method were also discussed in detail.

Comparisons of end-members of the size-shape distributions (convexity, circularity and aspect-ratio) with traditional grain size EMs also confirmed the accuracy of the novel approach. The remarkable similarity of size-aspect ratio and size EMs is quite surprising, especially because (as it was also noted by the authors) other studies (e.g., Shang et al. 2018) reported sorting of aspect ratio during transport, but in that case long-distance transport of silt-sized material was analysed (by using different magnification and aspect ratio definition).

All in all, the authors introduce successfully a new method and a new way of data processing of particle size and shape data. The new approach was effectively applied to determine major aeolian processes of the investigation area.

Minor comments:

Last columns of Table 2 are not visible in the manuscript.

The overall structure of the paper is good, however, figure from the Appendix A could be moved into the main text.

---

## Referee Comment (RC2) · Simon Blott (Referee) · 11 Feb 2020

This is a very good manuscript. It is well written, with good English, is extremely technical, although can be followed by a non-specialist. It is lengthy, but it is difficult to see how it could be substantially shortened without removing relevant material.

One small point is that Table 2 appears truncated and missing the right hand side. And it is also unfortunate that important material is in the Appendices at the back - I found myself referring to these throughout, which is a little tiresome moving back and forth to the end of the manuscript. I would recommend moving at least the first two appendices into the main text.

[Figure]

**ESurfD**

Interactive
comment

---

## Author Comment (AC2) · 26 Feb 2020

Dear Simon Blott,

Thank you for your positive comments and your suggestions. Below we have made a list of your comments (in quotation marks) and our replies.

Comment 1) "It is lengthy, but it is difficult to see how it could be substantially shortened without removing relevant material".

We also feel that, although the manuscript is long, there is not much opportunity for shortening it.

Comment 2) "One small point is that Table 2 appears truncated and missing the right hand side".

Thank you for noting this error. Part of the last column is indeed not visible in the manuscript. This will be changed in the revision.

Comment 3) "And it is also unfortunate that important material is in the Appendices at the back - I found myself referring to these throughout, which is a little tiresome moving back and forth to the end of the manuscript. I would recommend moving at least the first two appendices into the main text".

Do you suggest to move appendix A1 and A2 into the main text, or A and B? We will move A1, A2 and A3 into the main text, but appendix B will not be moved as it is of less importance.

---

## Author Response (AR1)

[revised manuscript text omitted]

**Response to the Anonymous Referee**

Dear anonymous referee,

Thank you for your positive review and valuable comments and suggestions.

Below you will find your comments and our replies in section 1. Section 2 contains a list of the changes that were made to the

5 manuscript.

**Section 1. Comments and replies**

Below we have made a list of your comments (in quotation marks and Italic) and our replies.

Comment 1) *"A more detailed description of the technique is suggested (Some suggested points:*

*Why are these methods better than previous sizing techniques? What are the advantages*

10 *and drawbacks compared to other imaging approaches? The repeated pumping*

*of the sample causes data redundancy [one particle will appear on more than one*

*captured frames]. Is it a problem, or not?)."*

1A) *"Why are these methods better than previous sizing techniques?"*

Many methods for measuring grain size have been/are employed, the most prevalent being sieving, settling and laser diffraction.

The first advantage over these methods is obvious: in addition to size analysis, image analysis allows measurement of grain shape, the traditional methods do not.

20 The second advantage is less obvious: the varying shape of natural sediment particles causes deviations in the size measurements obtained by sieving and laser diffraction: A sieve mesh actually measures the intermediate diameter of a particle and relatively large yet elongate particles can pass the mesh. With settling measurements, one has to make an assumption of the particle's shape (and density) to convert settling velocity to grain size.

In laser diffraction, the assumption is made that all particles are spherical. Since natural sediment grains are generally not

25 spherical, this causes errors in the obtained grain size distribution. Grain size obtained by image analysis is more robust in this sense: the (2D) shape of the particle is known, and therefore one can choose any definition of 'particle size'. For example, the largest diameter (2D), smallest diameter (2D) or 'average diameter'. We feel the latter is the most robust. For this reason, we used a diameter based on the surface area of the particle (D2d), as described in the manuscript. The third advantage of image analysis over traditional sizing techniques is that information is obtained on the size of each single grain passing the camera

30 rather than just the grain size distribution of the entire sample. This means 1) that single grains with a certain size can be extracted from the data for specific research questions. For example, one may want to know the largest grain per sample, 2) that single very large grains are detected (in our experience this is not the case with laser granulometry), 3) that grain size distributions can be made by percentage of the total volume of all grains per size class, but also by percentage of total diameter

or percentage of total number of particles, 4) that there are more options for statistical analysis of the sediment samples (which are as of yet not used).

There is however also a major drawback to the new technique: due to the pixel size of approximately 5 µm (or 2 µm with a different set-up which allows measurement of finer-grained particles but limits the maximum measurable grain size to 400 µm due to the 500 µm cuvette width), a reliable measurement of particle shape is not possible for anything finer than medium silt (Shang et al., 2018, Aeolian silt transport processes as fingerprinted by dynamic image analysis of the grain size and shape characteristics of Chinese loess and Red Clay deposits. Sedimentary geology, 375, 36-48). This is a financial, and not a technical limitation: the boundary could be lowered substantially with higher-resolution cameras.

1B) *"What are the advantages and drawbacks compared to other imaging approaches?"*.

The main advantage of dynamic image analysis (where particles pass a camera suspended in air or water) compared to the more classic static image analysis (where particles are photographed, for example under a microscope) lies in its statistical robustness and low measuring time: many particles can be measured in relatively little time (5 minutes for a typical number of particles of 150 thousand for sand (this manuscript) or 5 minutes for ~ a million particles for silts (Shang et al., 2018)). The large number of particles ensures that the size-shape distributions are statistically robust.

There are additional differences with the static methods: If the sample consists of unconsolidated sediment spread out on a flat surface, the orientation of the particles is known: particles lying on a surface have their smallest axis oriented vertically, and therefore their largest and intermediate axis show up on the image. This is both an advantage and disadvantage over our method: it is favourable that the orientation is known, but any inferences about the volume of the particles will always be overestimations of the actual volume.

Non-automated measurements using a microscope enable the computation of shape variables that are difficult to automate, such as surface texture of the grains or Power's roundness. However, such measurements are user-dependent, very time consuming and will comprise only a limited number of measured grains leading to less robust results.

1C) *"The repeated pumping of the sample causes data redundancy [one particle will appear on more than one captured frames]. Is it a problem, or not?"*

The most often used method for measuring grain size is laser diffraction. In most of the laser diffraction set-ups, the sample is repeatedly pumped through the cuvette as well. Therefore, this "problem" is universal in grain size measurements. In our view,

however, it is not a problem but rather something that adds to the strength of the measurement. Because the flow in the large cuvette is quasi-turbulent, the particles will pass the camera at different angles each time. The total measurement therefore becomes to some extent a 3D average of the various 2D shapes that one obtains by looking at a particle at different orientations. A size-shape distribution therefore becomes more robust then had the sediments passed the camera only once.

Comment 2) *"The description and the presented flow diagram ensure the reproducibility of the introduced measurement and data processing approach for experts of the field, but researchers without deeper knowledge on image analysis-based grain size and shape characterization may have trouble to understand the key steps of the method."*

10 We assume that this comment deals mainly with chapter 2.2 (dynamic image analysis). We enhanced apprehensibility of the chapter by adding a short description of the flow diagram (see section 2).

In our opinion, chapter 2.3 (construction and unmixing of size-shape distributions) and chapter 2.4 (artificial datasets for testing and validation of the method) go into sufficient detail to explain the method to non-experts.

Comment 3) *"The applied self-written Matlab script (with imfill, regionprops and convhull functions) is using the raw images of the acquired frames not processed data of the Sympatec Qicpic's software, allowing for a more detailed and freely customizable data handling. Are there any differences among the results of your own calculations (by using 'regionprops')*
20 *and ones by the device software?"*

Besides flexibility and customisability, there are two reasons for applying a self-written script:

3.1) The Qicpic software (our version at least) does not allow filling of blank spaces in particles. These blank spaces occur
25 because some minerals, such as quartz, are transparent, causing them to appear donut-shaped in the images. Due to our quartz-rich samples, we were not able to use any of the built-in functions for computing area-based shape parameters. Matlab, on the other hand, has built-in functions to perform the task of filling blank spaces in objects. At the moment, however, we use the perimeter to determine the 2D surface area of the particles and therefore blank spaces are less of a problem.

30 3.2) The Qicpic software (and many of Matlab's built-in functions such as regionprops('ConvexHull')) consider pixels to have an area (a pixel is a little square). In our method, we base computations on the pixel's centre location.

Comment 4) *"The presented size-shape distributions are equivalent to volume-weighted scatter plots of size and different shape parameters of individual particles. (It is a relatively well*

*known approach of image analysis-based granulometric characterization, actually, a*

*default data visualization mode in the software of Malvern Morphologi automated static*

*image analyser device)."*

5    We were not aware that grain size-shape distributions are a well-known approach. The size-shape distributions or volume-weighted scatter plots of size and different shape parameters can indeed  be found in Malvern's documentation: https://www.cif.iastate.edu/sites/default/files/uploads/Other_Inst/Particle%20Size/Particle%20Characterization%20Guide.pdf. Furthermore, we found one application in powder metallurgy: Takashi Itoh & Yoshimoto Wanibe, 1991: Particle Shape Distribution and

10    Particle Size–Shape Dispersion Diagram, Powder Metallurgy, 34:2, 126-134, DOI: 10.1179/ pom.1991.34.2.126. However, we found no applications of a similar method in sedimentology. Thus, grain size-shape distributions are not new, but their application to the study of sediments is new. More importantly, the combination with end-member modelling is also new. We added the reference of Itoh & Wanibe to the manuscript (see section 2).

15    Comment 5) *"Last columns of Table 2 are not visible in the manuscript."*

Thank you for noting this error. Part of the last column is indeed not visible in the manuscript. This is changed in the revision.

Comment 6) *"The overall structure of the paper is good, however, figure from the Appendix A could*

20    *be moved into the main text."*

Appendix A is moved into the main text of the revision.

10 1A) *"Why are these methods better than previous sizing techniques?"*

A comparison of dynamic image analysis to other sizing techniques is outside the scope of this manuscript. The full reply in section 1 was therefore not taken up into the manuscript.

However, we have included a brief explanation of the advantage of knowing the two-dimensional shape of a particle when
15 determining its size (page 6, line 10 to 14). We deleted: *"For the same reason, "the" diameter of the particle is given in the robust form of an area equivalent diameter (Table 2)."* And we added: *""The" grain size of the particle is given in the form of an area equivalent diameter (Table 2), essentially the average particle diameter of the two-dimensional image of the grain. Because the two-dimensional shape of the particle is known, grain size obtained by image analysis is more robust than traditional size measurements (e.g. sieving, laser diffraction and settling) where an assumption has to be made of particle
20 shape before computing size (Konert and Vandenberghe, 1997)."*

1B *"What are the advantages and drawbacks compared to other imaging approaches?".*

A detailed comparison of dynamic image analysis to other imaging approaches is also outside the scope of this manuscript.
25 We therefore did not include our answer into the revised manuscript.

1C) *"The repeated pumping of the sample causes data redundancy [one particle will appear on more than one captured frames]. Is it a problem, or not?"*

30 The answer to 1C) was not included in the manuscript because repeated pumping of a sample is very common in grain size measurements and therefore not new to our method.

Comment 2) *"The description and the presented flow diagram ensure the reproducibility of the introduced measurement and data processing approach for experts of the field, but researchers without deeper knowledge on image analysis-based grain size and shape characterization may have trouble to understand the key steps of the method."*

We enhanced apprehensibility of the chapter by adding a short description of the flow diagram (page 6, line 4 to 8):

*"In the first step of the script some limitations and conditions are set. Subsequently, the script iterates over each video, over each frame in the video and over each particle found in the frames. For each particle, the length of its outer edge (perimeter) is computed, as well as its area and the length of its convex hull (a polygon drawn around the particle without taking into*
5   *account the concave areas). These basic parameters are stored for each particle. Particle size, volume, aspect ratio, convexity and Cox circularity are subsequently computed from these basic parameters."*

Comment 3) *"The applied self-written Matlab script (with imfill, regionprops and convhull functions) is using the raw images of the acquired frames not processed data of the Sympatec Qicpic's software, allowing for a more detailed and freely*
10   *customizable data handling. Are there any differences among the results of your own calculations (by using 'regionprops') and ones by the device software?"*

A discussion on the differences between the Sympatec software, the built-in Matlab functionality and our functions is outside the scope of the manuscript. However, the fact that we fill blank spaces in the particles and that we compute size and shape by
15   the pixel's centre locations are important for reproducibility of this work. This information was already present in the workflow diagram of the script (Appendix B, which was appendix C in the previous version of the manuscript)

Comment 4) *"The presented size-shape distributions are equivalent to volume-weighted scatter plots of size and different shape parameters of individual particles. (It is a relatively well known approach of image analysis-based granulometric*
20   *characterization, actually, a default data visualization mode in the software of Malvern Morphologi automated static image analyser device)."*

We changed: "In this study we outline a new method for determination of sediment transport processes involving 1) the integration of grain size and shape data into size-shape distributions and 2) end-member modelling on these distributions."
25   (Introduction, page 3, lines 8-10) to: *"In this study we outline a new method for determination of sediment transport processes involving 1) the integration of grain size and shape data into size-shape distributions (e.g. Itoh and Wanibe, 1991) and 2) end-member modelling on these distributions".*

Comment 5) *"Last columns of Table 2 are not visible in the manuscript."*
30
Table 2 was changed in the revised manuscript to include the last column.

Comment 6) *"The overall structure of the paper is good, however, figure from the Appendix A could be moved into the main text."*

Appendix A has been moved into the main text as Fig. 1.

**Response to Simon Blott**

Dear Simon Blott,

Thank you for your positive comments and your suggestions.

5    Section 1 contains our replies to your comments. Section 2 describes the changes made to the manuscript based on your comments.

**Section 1**

Below we have made a list of your comments (in italic and quotation marks) and our replies.

10    Comment 1) *"It is lengthy, but it is difficult to see how it could be substantially shortened without removing relevant material"*.
We also feel that, although the manuscript is long, there is not much opportunity for shortening it.
Comment 2) *"One small point is that Table 2 appears truncated and missing the right hand side"*.
Thank you for noting this error. Part of the last column is indeed not visible in the manuscript. This is changed in the revision.
Comment 3) *"And it is also unfortunate that important material is in the Appendices at the back - I found myself referring to*

15    *these throughout, which is a little tiresome moving back and forth to the end of the manuscript. I would recommend moving at least the first two appendices into the main text"*.
Do you suggest to move appendix A1 and A2 into the main text, or A and B? We moved A1, A2 and A3 into the main text, but appendix B is not moved as it is of less importance.

**Section 2**

20    Below we have made a list of your comments and the accompanying changes we made to the manuscript.
Comment 1) *"It is lengthy, but it is difficult to see how it could be substantially shortened without removing relevant material"*.
We did not shorten the manuscript.
Comment 2) *"One small point is that Table 2 appears truncated and missing the right hand side"*.
Table 2 was updated to include the last column.

25    Comment 3) *"And it is also unfortunate that important material is in the Appendices at the back - I found myself referring to these throughout, which is a little tiresome moving back and forth to the end of the manuscript. I would recommend moving at least the first two appendices into the main text"*.
We moved appendix A1, A2 and A3 into the main text as figure 1.

---

## Author Response (AR2)

[revised manuscript text omitted]

**Dear Niels Hovius and Patricia Wiberg,**

Thank you for helping improve the manuscript.

Replies to your comments can be found in section 1, insofar these comments require a reply. Section 2 contains a list of the changes that were made to the manuscript with regard to your comments.

5 **Section 1: comments and replies**

*Comment 1: "The application of the method, much of the general interpretation of end members and grain shape, and most of the citations focus on aeolian transport. Has the method been applied to characterize subaqueous transport? I think the text (and even the title and abstract) should make it clearer that the focus of the development and application of the method presented in the paper is related to aeolian transport.*

10 This method was applied to an aeolian sedimentary system. However, the method can be applied to other sedimentary environments as well (the main difference between saltation, modified saltation and suspension transport in water versus air is the density and viscosity of the medium, and hence sorting in size and shape will be similar in direction, not necessarily in quantity). In fact, the method has by now been applied to subaqueous transport, but this is to be published in another manuscript. We understand that this might lead to confusion

15 and have therefore changed several parts of the text, including the title (see section 2) to better reflect the application to an aeolian system in the manuscript.

In the sections with artificial data, we emphasized that the end members were sourced from the aeolian dune dataset. No further changes were made to those sections of the manuscript because any type of sediment, or even made-up size-shape distribution end-members could have been used for the artificial datasets with the

20 same effect. Aeolian size-shape distributions were simply what we had on hand to put the method through its paces.

The first section of the introduction (lines 1-22) was also not changed, because it serves as a general introduction to the problem of end-member modelling on one variable (size). Both aeolian and non-aeolian references were used here.

25 *Comment 7) "P15, L25: Legend for Appendix I (formerly J) still refers to panels as J1, J2, as does the figure itself. Also on L24 I would replace word "drastically" with something like "greatly"".*

We are very sorry this mistake was not fixed in the last revision. Apparently the old rather than the new version of the figure got into the manuscript.

30 **Section 2: comments and changes**

*Comment 1) "The application of the method, much of the general interpretation of end members and grain shape, and most of the citations focus on aeolian transport. Has the method been applied to characterize*

*subaqueous transport? I think the text (and even the title and abstract) should make it clearer that the focus of the development and application of the method presented in the paper is related to aeolian transport.*

Page 1, line 20: removed "also" because it may lead to the suspicion that the method was also tested on other non-aeolian systems which is not the case for the current study.

Page 1, line 22: added "aeolian" to the name of the dune dataset to remove ambiguity about the sedimentary environment referred to.

Page 1, line 30: added "aeolian" to reflect the focus of the current study on the aeolian system.

Page 2, line 24-25: added "The role of particle shape in aeolian transport is highlighted because the method presented in the current paper is tested on an aeolian system." to better reflect the focus of this section on aeolian transport.

Chapter 2.1:

Page 3, line 18 & page 5, line 10: added "aeolian" to the name of the dune dataset.

Page 3, line 30: added "aeolian"

Chapter 2.2 &2.3: no changes were made because dynamic image analysis, construction of size-shape distributions and end-member modelling of size-shape distributions can be carried out on any type of sediment.

Chapter 2.4:

Page 9, line 15: added "aeolian" to the name of the dune dataset to avoid ambiguity concerning the source of the end members used for the artificial datasets.

Chapter 3.2:

Page 15, line 3 and 5 & page 16, line 16 & page 17, line 6 & page 18, line 2 & page 19, line 12: added "aeolian" to the name of the dune dataset.

Page 15, line 6-7: changed "statistics for the ConD2d dataset are shown first to derive the number of end members necessary to explain size-shape variability of the dataset" to "statistics for the ConD2d dataset are shown first to derive the number of end members necessary to explain grain size-shape variability occurring in the aeolian deposits". This change was made to remind the reader that the end member modelling is applied to infer the primary sedimentary components in an aeolian system.

Chapter 4.1:

Page 21, line 12,14,16,20: added "aeolian" to the name of the dataset.

Page 21, line 19: changed "processes" to "aeolian processes".

Chapter 4.2:

Page 21, line 29: added "aeolian" to the name of the dataset.

Page 23, line 25: changed "transport modes" to "aeolian transport modes"

Page 23, line 16-17: removed "during transport" from line 16, and added "during aeolian transport" to line 17.

Chapter 5:

Page 24, line 2: changed "sedimentary deposits" to "aeolian deposits" to highlight the focus of this manuscript on aeolian processes.

Page 24, line 15: added "aeolian" to the name of the dune dataset.

*Comment 2) "Fig. 1B: labels for field sites are very small. Perhaps coding the 4 regions by color and just numbering each with a larger number would make it easier to determine which site, e.g., is A1 or C3".*

Numbers of the sediment trap locations where enlarged. Font size of legends has also been enlarged.

*Comment 3) "P5, L5&7: Rather than "land inward from" or land inward area" I suggest using "landward from" and "landward area" or "inland area".*

*These changes have been made.*

*Comment 4) "P6, L11: It would be helpful to add the abbreviation for the area equivalent diameter. That is, "in the form of an area equivalent diameter (D2d, Table 2)".*

*This change has been made.*

*Comment 5) "Table 2: I wonder if there is a way to rearrange this so that the first 5 entries are not listed with no information in the last 6 columns. Perhaps just have the derived characteristics in the table with the symbol, name and description of the measured variables in a separate table or in the table heading".*

The basic particle characteristics were removed from the table.

The following was deleted from the table header: "characteristics and derived" (because the basic particle characteristics were removed from the table).

The following was added to the table header: "The following letters were used in the equations for the size-shape variables: Pp (perimeter, or length along the particle boundary), Pch (convex hull or length along the convex points on the particle boundary), A (particle area), DB (minor diameter of ellipse fitted to particle), DA (corresponding major diameter)".

*Comment 6) "P15, L10: It seems awkward to begin a section with a reference to an appendix. Consider adding the figure in Appendix H to the text; or perhaps omit the first sentence and just refer to Appendix H at the end of the next sentence.*

The reference to appendix H was moved to the end of sentence two. Sentence one (line 10-11) was changed from "Appendix H displays the trend of median class-wise and sample-wise R2 against the number of end members" to "The trend of median class-wise and sample-wise R2 against the number of end members can be used as a primary indication of the number of primary components necessary for a good representation of the aeolian dune dataset."

*Comment 7) "P15, L25: Legend for Appendix I (formerly J) still refers to panels as J1, J2, as does the figure itself. Also on L24 I would replace word "drastically" with something like "greatly"".*

The new figure has now been included in the manuscript. The text for appendix I has been changed accordingly. Furthermore, we noted an erroneous reference to appendix A3 (page 31, line 11) which is now Fig. 1C. The same mistake was found (and changed) on page 18 L5, page 20 L3, page 34, L4, page 35, L6

*Comment 8) "P19, Fig. 12: The colors corresponding to EM1-3 and EM3-3 are reversed and need to be corrected".*

Thank you for noting. We reversed the colours.

*Comment 9) "P22, L15: "Sediment traps on the…"*

This mistake was fixed.